# Test-Time Guidance for Flow-Based Generative Models
# via Parallel Tempering on Source Distributions

**Shih-Hsin Wang** [* 1]  **Joel A. Keller** [* 1]  **Taos Transue** [1]  **Drake Benjamin Brown** [1]  **Thomas Strohmer** [2]  **Bao Wang** [1]

## Abstract

Generative models that transport a simple source distribution to a complex data distribution—such as diffusion and flow-based models—are central to high-fidelity data generation. Test-time guidance can further steer pretrained models toward user-specified high-reward regions without costly retraining. However, existing guidance methods face critical limitations: they struggle with non-differentiable rewards, fail to navigate complex landscapes, and often lack theoretical guarantees on generation performance. We propose *Source Parallel Tempering (SPT)*, a gradient-free test-time guidance framework that operates entirely in source space, leveraging its simpler geometry to avoid the complexities of the data manifold. SPT couples a local exploration kernel with parallel tempering, enabling efficient barrier crossing and robust discovery of high-reward modes. Theoretically, we provide a new error bound linking training-time approximation error to test-time guidance performance. Empirically, SPT significantly improves over state-of-the-art methods on benchmark tasks in conditional image synthesis and dynamical system trajectory sampling. Code is available at https://github.com/Utah-Math-Data-Science/SPT.

## 1. Introduction

Generative models (GMs) that transport a simple source distribution to a complex data distribution—such as diffusion models (DMs) [40; 21; 43] and flow-based models [29; 2; 30]—have become the cornerstone of high-fidelity

---
[*]Equal contribution  [1]Department of Mathematics and Scientific Computing Institute, University of Utah, Salt Lake City, UT 84102, USA [2]Department of Mathematics, University of California, Davis, California 95616, USA. Correspondence to: Bao Wang <wangbaonj@gmail.com>.

*Proceedings of the 43rd International Conference on Machine Learning*, Seoul, South Korea. PMLR 306, 2026. Copyright 2026 by the author(s).

data synthesis. Despite their success, practical deployment often necessitates *guided generation*, where outputs must adhere to strict semantic or physical constraints [20; 39]. While training-time guidance (e.g., classifier-free guidance [20]) offers good performance, it lacks flexibility, as any new constraint necessitates model retraining. This limitation has motivated the development of *test-time guidance*, which seeks to directly sample from a *reward-tilted distribution*:

$$\tilde{p}(\boldsymbol{x}) \propto p(\boldsymbol{x}) \exp(\beta R(\boldsymbol{x})) \tag{1}$$

using a fixed pretrained GM [27] that was originally trained to produce samples from data distribution $p(\boldsymbol{x})$. Here, $R(\boldsymbol{x})$ represents a task-specific reward function that encodes the desired properties, and $\beta > 0$ is the inverse temperature parameter that controls the strength of conditioning.

However, developing effective test-time guidance methods remains an open challenge. Gradient-based steering methods [43; 4] fail when $R(\boldsymbol{x})$ is non-differentiable, such as binary success or failure signals in control tasks, or when gradients are computationally intractable. To address this, recent works have introduced *gradient-free* steering methods, including Feynman-Kac (FK) Steering [39], CREPE [19], and Source Guidance [47]. While promising, these approaches face two significant and unresolved challenges. First, these methods often struggle to balance high reward and high probability. For example, approaches such as FK Steering fail to identify high-reward regions when they reside in low-density areas. As illustrated by our numerical results in Fig. 1, this limitation becomes acute as the reward width parameter $\omega^2$ decreases, creating an increasingly sharp global optimum in a low-density region. Specifically, as the peak narrows, FK Steering remains trapped in the local optima corresponding to the original data mass, leading to a significant increase in the Wasserstein distance [45] between the sampled and target distributions.

Second, there is a lack of end-to-end theoretical guarantees on how approximation or training errors in the underlying GM propagate to the quality of the guided samples. Existing analyses often assume exact flows [39] or unrealistic $L_\infty$ error bounds [47], leaving open how the standard $L_2$ training objectives—used in e.g., flow matching (FM) [29; 2; 30; 24]—translate to the generation quality of guided

frameworks.

## 1.1. Our Contribution

To address these limitations, we propose *Source Parallel Tempering (SPT)*, a gradient-free guidance framework designed to sample effectively from highly multimodal reward landscapes (see Section 4.1). By steering candidates in the source space, SPT circumvents treating the complex data distribution, enabling it to utilize two complementary mechanisms to ensure high-dimensional scalability: parallel tempering (PT) [12] coupled with the preconditioned Crank–Nicolson (pCN) operator [5]. The pCN operator acts as a gradient-free local kernel with dimension-independent convergence rates. Simultaneously, PT facilitates global exploration by maintaining a ladder of Markov chains at varying inverse temperatures from 0 to $\beta$. By periodically swapping states between these chains, SPT allows samples to traverse low-probability barriers and escape local traps, leading to the discovery of diverse high-reward modes [32].

Theoretically, we provide a new analysis bridging training-time approximation error and test-time generation performance. In particular, we establish a novel end-to-end error bound for test-time guidance, showing that the standard $L_2$ approximation error of a pretrained GM directly controls the 2-Wasserstein distance between the distribution of samples generated by SPT and the reward-tilted distribution $\tilde{p}$ (Corollary 4.4). This result justifies the use of PT with source steering in test-time guidance and clarifies how model training quality governs guidance performance.

We validate the effectiveness of SPT across diverse benchmark tasks, including image synthesis and dynamical system trajectory sampling (see Section 5). Notably, the dynamical system task in 5.3 employs a non-differentiable binary reward, serving as a critical stress test for the gradient-free capabilities of our method. Empirically, SPT achieves substantial improvements over baseline methods, with gains of up to 50% in both reward and standard sample quality metrics, highlighting its ability to navigate complex, multimodal reward landscapes and generate high-reward samples without compromising sample quality across domains.

## 1.2. Additional Related Works

While PT has been explored in GMs—e.g., CREPE [19] and Markovian Flow Matching [9]—these approaches typically rely on time-based tempering and gradient-based Langevin dynamics. Our method, SPT, differs fundamentally by applying *reward-based* tempering within the source space, enabling compatibility with fast ODE solvers. We also distinguish our work from recent source steering methods [47]. While Wang et al. [47] introduce a source steering framework, their proposed sampling algorithms are either restricted to gradient-based settings or limited to low dimen-

sions. Furthermore, as detailed in Section 3, their theoretical guarantees rely on impractical $L_\infty$ norm assumptions. In contrast, we establish a new end-to-end performance guarantee that holds for FMs and DMs.

## 1.3. Organization

We organize this paper as follows: Section 2 reviews background on flow-based generative models and existing test-time guidance methods. Sections 3 and 4 introduces the SPT framework, its algorithmic components, and its performance guarantees. Section 5 reports empirical results for image generation and dynamical system benchmarks. This paper concludes with limitations and future directions.

## 2. Background: Flow-based GMs

Flow-based GMs stand out as a class of efficient and flexible models for generative modeling. These models define a flow induced by an ODE $d\boldsymbol{x}_t = \boldsymbol{v}_t(\boldsymbol{x}_t)\,dt$, transporting a simple source distribution $q(\boldsymbol{z})$ (e.g. $\mathcal{N}(\boldsymbol{0}, \boldsymbol{I})$) at $t = 0$ to a target data distribution $p(\boldsymbol{x})$ at $t = 1$. Below, we discuss several prominent formulations.

In score-based GMs [41; 43], the velocity is given by $\boldsymbol{v}_t(\boldsymbol{x}) = \boldsymbol{f}(\boldsymbol{x}, t) - \frac{1}{2}\boldsymbol{g}^2(t)\nabla_{\boldsymbol{x}}\log p_t(\boldsymbol{x})$, where $p_t(\boldsymbol{x})$ denotes the intermediate marginal density. In training, the score $\nabla_{\boldsymbol{x}}\log p_t(\boldsymbol{x})$ is approximated by a neural network $\boldsymbol{s}_\theta(t, \boldsymbol{x})$, yielding an estimated velocity $\boldsymbol{v}_\theta(t, \boldsymbol{x})$.

Flow matching [29; 2; 30] instead directly parameterizes $\boldsymbol{v}_t$ as a neural network $\boldsymbol{v}_\theta(t, \boldsymbol{x})$ and learns it via minimizing the following simulation-free regression objective:

$$\mathcal{L}(\theta) = \mathbb{E}_{t,\boldsymbol{x}_0,\boldsymbol{x}_1}\big\|\boldsymbol{v}_\theta(t, \boldsymbol{I}_t) - \partial_t \boldsymbol{I}_t\big\|^2, \qquad (2)$$

where $\boldsymbol{I}_t$ represents an interpolation between $\boldsymbol{x}_0 \sim q$ and $\boldsymbol{x}_1 \sim p$. Crucially, the unique global minimizer of this objective satisfies $\boldsymbol{v}_t(\boldsymbol{x}) = \mathbb{E}_{\boldsymbol{x}_0,\boldsymbol{x}_1}[\partial_t \boldsymbol{I}_t \mid \boldsymbol{I}_t = \boldsymbol{x}]$.

In both frameworks, the velocity field $\boldsymbol{v}_t(\boldsymbol{x})$ induces a deterministic transport map $T$ defined by the integral:

$$T(\boldsymbol{x}_0) = \boldsymbol{x}_0 + \int_0^1 \boldsymbol{v}_t(\boldsymbol{x}_t), dt. \qquad (3)$$

The approximate transport map $T_\theta$ is obtained by substituting the true velocity field $\boldsymbol{v}_t$ with its approximation $\boldsymbol{v}_\theta$.

It has been shown in [2] that the gap between $T$ and $T_\theta$ is upper-bounded by the training loss in learning the vector field. In particular, we have:

$$\begin{aligned}
&\mathbb{E}_{\boldsymbol{z}\sim q}\big[\|T(\boldsymbol{z}) - T_\theta(\boldsymbol{z})\|^2\big] \\
&\leq e^{1+2L_{\boldsymbol{v}_\theta}} \cdot \mathbb{E}_{t,\boldsymbol{x}_0,\boldsymbol{x}_1}\big\|\boldsymbol{v}_\theta(t, \boldsymbol{I}_t) - \boldsymbol{v}_t(\boldsymbol{I}_t)\big\|^2,
\end{aligned} \qquad (4)$$

where $L_{\boldsymbol{v}_\theta}$ denotes the Lipschitz constant of the approximated velocity field $\boldsymbol{v}_\theta(t, \boldsymbol{x})$. This result ensures that minimizing $\mathcal{L}(\theta)$ in equation 2 controls the generation error.

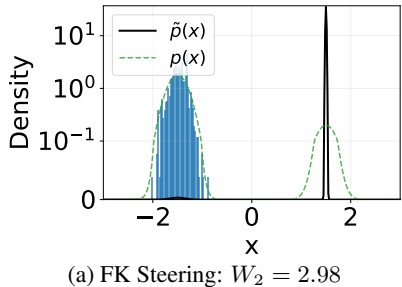
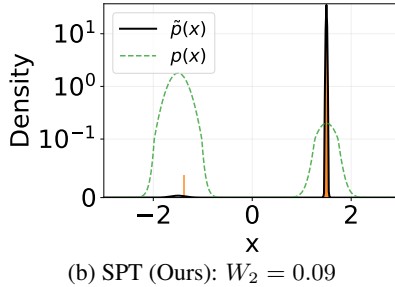
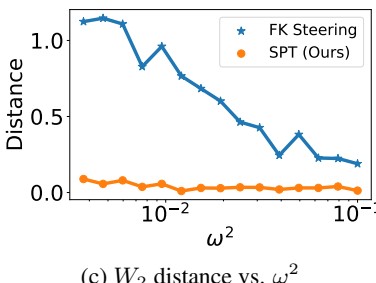

(a) FK Steering: $W_2 = 2.98$    (b) SPT (Ours): $W_2 = 0.09$    (c) $W_2$ distance vs. $\omega^2$

*Figure 1.* Panel (a) shows a sample where FK Steering fails to capture the high-reward mode of the tilted distribution $\tilde{p}(x)$ in equation 14 for $\omega^2 = 0.01$, missing the spiky region near $x = 1.5$. Panel (b) shows a separate sample from our method, which successfully captures the correct mode, illustrating its effectiveness on sharp distributions. Panel (c) quantifies this behavior over 50 independent trials: the 2-Wasserstein distance between the FK-sampled and exact distributions spikes as $\omega^2$ decreases. In contrast, our method remains stable, yielding smooth and accurate approximations regardless of the peak width. See Section 5.1 for experimental details.

Instead of learning $\boldsymbol{v}_\theta(t, \boldsymbol{x})$, another popular paradigm involves directly learning the transport map $T_\theta$ to enable single-step generation. Prominent examples include consistency models (CM) [44; 42], consistency trajectory models (CTM) [25], MeanFlow [16], and flow maps [8; 7]. Theoretical guarantees similar to equation 4 for these direct mapping methods have been established in [8].

## 3. Source Distribution Steering

Before detailing our algorithm, we first establish the theoretical foundations of source-space guidance. We begin by reviewing the fundamental duality principle that enables steering samples in the source space. Importantly, we provide a novel analysis connecting the GM's training objective—measured as an $L_2$ approximation error—to test-time generation quality (Theorem 3.3). This addresses a key limitation of prior work [47], which relied on $L_\infty$ pointwise error bounds that are not directly controlled by standard training objectives.

### 3.1. Equivalence of Source and Data Steering

Consider a GM that defines a transport map $T$ pushing a simple source distribution $q(\boldsymbol{z})$ to a complex data distribution $p(\boldsymbol{x})$. Recent work [47] introduces a test-time guidance framework that operates entirely within the source space, distinct from methods that steer in data or intermediate spaces. This approach is grounded in a fundamental duality principle: instead of steering the complex data distribution $p(\boldsymbol{x})$ toward regions of high reward $R(\boldsymbol{x})$, one can equivalently steer the source distribution $q(\boldsymbol{z})$ toward high-reward regions $R(T(\boldsymbol{z}))$ induced by the map $T$.

Formally, let $T : \mathcal{Z} \to \mathcal{X}$ be a transport map that pushes $q(\boldsymbol{z})$ to $p(\boldsymbol{x})$, denoted by $p = [T]_\sharp q$. By the change of variables formula [26], the densities satisfy

$$p(\boldsymbol{x}) = q(\boldsymbol{z}) \left| \det \nabla_{\boldsymbol{z}} T(\boldsymbol{z}) \right|^{-1}, \text{ where } \boldsymbol{x} = T(\boldsymbol{z}). \quad (5)$$

We aim to sample from the tilted target distribution in equation 1, leveraging the *tilted source distribution*:

$$\tilde{q}(\boldsymbol{z}) \propto q(\boldsymbol{z}) e^{\beta R(T(\boldsymbol{z}))}. \quad (6)$$

The following proposition shows that generating samples by transporting $\tilde{q}$ is equivalent to sampling directly from $\tilde{p}$.

**Proposition 3.1** (Wang et al. [47, Theorem 1])**.** *The pushforward of the tilted source distribution $\tilde{q}$ under the map $T$ is exactly the tilted target distribution $\tilde{p}$. That is, $\tilde{p} = [T]_\sharp \tilde{q}$.*

### 3.2. Robustness to Transport Map Approximation

In practice, the true transport map $T$ is unknown; GMs instead learn an approximation $T_\theta$ by minimizing a training objective summarized in Section 2. This training procedure ensures that $T_\theta$ is close to $T$ in the $L_2$ sense, yielding a bound on the approximation error $\epsilon^2 = \mathbb{E}_{\boldsymbol{z} \sim q} \left[ \| T(\boldsymbol{z}) - T_\theta(\boldsymbol{z}) \|^2 \right]$. However, this naturally raises a fundamental question regarding the reliability of source steering:

*Is the generation error between the true tilted target $\tilde{p}$ and the tilted target distribution induced by the approximate map $T_\theta$ bounded by the approximation error $\epsilon$?*

Answering this question is essential for connecting optimization at training time to guarantees on test-time guided generation quality, effectively justifying the use of approximate transport maps for source steering.

The authors of [47] investigated this question by assuming a uniform bound on the vector field approximation error:

**Theorem 3.2** (Wang et al. [47, Theorem 2])**.** *Assume that*

$$\left\| \boldsymbol{v}_t(\boldsymbol{x}) - \boldsymbol{v}_t^\theta(\boldsymbol{x}) \right\|_\infty \le \epsilon,$$

*and the learned velocity field $\boldsymbol{v}_t^\theta(\boldsymbol{x})$ is $L_{\boldsymbol{v}}$-Lipschitz continuous in $\boldsymbol{x}$. Suppose that the sampling method returns samples of distribution $\tilde{q}_{\text{sample}}$. Then, we have*

$$W_2 \left( \tilde{p}, [T_\theta]_\sharp \tilde{q}_{\text{sample}} \right) \le e^{L_{\boldsymbol{v}}} W_2 \left( \tilde{q}, \tilde{q}_{\text{sample}} \right) + \epsilon e^{L_{\boldsymbol{v}}}. \quad (7)$$

A fundamental limitation of this result is its reliance on the $L_\infty$ (worst-case) error bound. In practice, standard training objectives for FMs and DMs minimize an $L_2$-averaged error between vector fields, which does not guarantee the pointwise uniformity assumed by Theorem 3.2. This creates a gap between theory and practice: existing bounds fail to justify the high-quality guidance observed from models trained with standard $L_2$ losses.

To bridge this gap, we establish a theoretical characterization based directly on the $L_2$ norm, yielding bounds that are both practically relevant and theoretically consistent with standard training objectives. Specifically, the following theorem demonstrates that the 2-Wasserstein generation error [45] scales linearly with the map approximation error.

**Theorem 3.3** (Informal). *Let* $\tilde{q}_\theta(z) \propto q(z)e^{\beta R(T_\theta(z))}$ *be the approximate tilted source distribution. Let* $\epsilon^2 = \mathbb{E}_{z \sim q}[\|T(z) - T_\theta(z)\|^2]$ *denote the expected squared error of the learned map, which is upper bounded by the training loss used to fit* $T_\theta$ *(see equation 4 in Section 2). Under mild regularity conditions, the generated distribution* $[T_\theta]_\sharp \tilde{q}_\theta$ *remains close to the true tilted target* $\tilde{p}$; *in particular, there is a constant* $C > 0$ *s.t.*

$$W_2\left(\tilde{p}, [T_\theta]_\sharp \tilde{q}_\theta\right) \leq C \cdot \epsilon. \tag{8}$$

We provide the formal version together with the proof of this theorem, including detailed assumptions—standard in the literature or satisfied in practice—in Appendix A. Crucially, this result guarantees that as $\epsilon \to 0$, sampling via source steering asymptotically recovers the target tilted distribution.

## 4. Source Parallel Tempering

### 4.1. Algorithm

Building on the theory established above, we introduce SPT (see Algorithm 1)—a principled adaptation of the classical PT framework for test-time guidance. Crucially, SPT operates by steering a Gaussian source distribution $q = \mathcal{N}(\mathbf{0}, \mathbf{I})$, which we assume throughout, to sample from the complex target distribution $\tilde{q}_\theta$. It maintains parallel Markov chains at different temperatures of the reward constraint, exploiting the simple geometry of the source space to cross low-probability energy barriers and efficiently discover and refine high-reward modes. Now, we lay out SPT in detail.

**1. Temperature Ladder.** We construct a sequence of target distributions $\{\pi_k\}_{k=0}^K$ that interpolate between the original source $q = \mathcal{N}(\mathbf{0}, \mathbf{I})$ and the tilted source $\tilde{q}_\theta$. Defining a ladder of inverse temperatures $0 = \beta_0 < \beta_1 < \cdots < \beta_K = \beta$, the density for the $k$-th chain is:

$$\pi_k(z) \propto q(z) \exp(\beta_k R(T_\theta(z))). \tag{9}$$

This stratification bridges the easily explorable prior ($\pi_0 = q$) and the complex, highly constrained target ($\pi_K = \tilde{q}_\theta$).

---

**Algorithm 1** Source Parallel Tempering (SPT)

1: **Input:** Pre-trained transport map $T_\theta$, Reward $R(\boldsymbol{x})$, Ladder sizes $K$, Steps $t_{\max}$.
2: **Initialize:** Inverse temperatures $0 = \beta_0 < \cdots < \beta_K = \beta$. Step sizes $\theta_0 > \cdots > \theta_K$. Particles $\boldsymbol{z}_0^{(k)} \sim \mathcal{N}(\mathbf{0}, \boldsymbol{I})$ for $k = 0, \ldots, K$.
3: **for** $t = 1$ to $t_{\max}$ **do**
4:     *# 1. Local Exploration*
5:     **for** $k = 0$ to $K$ **do**
6:        Sample noise $\boldsymbol{\xi} \sim \mathcal{N}(\mathbf{0}, \boldsymbol{I})$.
7:        Propose $\tilde{\boldsymbol{z}} = \cos(\theta_k)\boldsymbol{z}_{t-1}^{(k)} + \sin(\theta_k)\boldsymbol{\xi}$.
8:        Compute $\Delta E = \beta_k[R(T_\theta(\tilde{\boldsymbol{z}})) - R(T_\theta(\boldsymbol{z}_{t-1}^{(k)}))]$.
9:        Calculate acceptance $\alpha = \min\left(1, \exp(\Delta E)\right)$.
10:       Sample $u \sim \text{Uniform}[0, 1]$.
11:       **if** $u < \alpha$ **then**
12:         $\boldsymbol{z}_t^{(k)} \leftarrow \tilde{\boldsymbol{z}}$ {Accept}
13:       **else**
14:         $\boldsymbol{z}_t^{(k)} \leftarrow \boldsymbol{z}_{t-1}^{(k)}$ {Reject}
15:       **end if**
16:     **end for**
17:     *# 2. Replica Exchange*
18:     **for** $k = 0$ to $K - 1$ **do**
19:       Compute $\Delta E_{\text{swap}} = (\beta_{k+1} - \beta_k)\big[R(T_\theta(\boldsymbol{z}_t^{(k)})) - R(T_\theta(\boldsymbol{z}_t^{(k+1)}))\big]$.
20:       $\alpha_{\text{swap}} = \min(1, \exp(\Delta E_{\text{swap}}))$.
21:       **if** $u < \alpha_{\text{swap}}$ **then**
22:         Swap states: $\boldsymbol{z}_t^{(k)} \leftrightarrow \boldsymbol{z}_t^{(k+1)}$.
23:       **end if**
24:     **end for**
25: **end for**
26: **Output:** The final data sample $\boldsymbol{x}_{\text{SPT}} = T_\theta(\boldsymbol{z}_{t_{\max}}^{(K)})$.

---

**2. Local Exploration.** We use the pCN operator [5], which yields **dimension-independent** acceptance rates under a Gaussian prior. We assign a decreasing sequence of step sizes $\frac{\pi}{2} \approx \theta_0 > \cdots > \theta_K > 0$, enabling a transition from global exploration to fine-grained refinement. At iteration $t$, the proposal $\tilde{\boldsymbol{z}}$ for chain $k$ is generated via:

$$\tilde{\boldsymbol{z}} = \cos\theta_k\, \boldsymbol{z}_{t-1}^{(k)} + \sin\theta_k\, \boldsymbol{\xi}, \quad \boldsymbol{\xi} \sim \mathcal{N}(\mathbf{0}, \boldsymbol{I}). \tag{10}$$

*Rationale:* A key property of the pCN operator is that its transition kernel satisfies *detailed balance* with respect to the prior $q(\boldsymbol{z})$. Consequently, the prior terms cancel in the Metropolis–Hastings ratio, simplifying the acceptance probability to depend solely on the reward improvement:

$$\alpha = \min\left(1, \exp(\beta_k[R(T_\theta(\tilde{\boldsymbol{z}})) - R(T_\theta(\boldsymbol{z}_{t-1}^{(k)}))])\right).$$

Crucially, the rotation angle $\theta_k$ is explicitly calibrated to the chain index $k$ to balance exploration and acceptance. For lower chains (small $\beta_k$), we select a larger $\theta_k$ to encourage aggressive, global jumps across the source space.

Conversely, for higher chains (large $\beta_k$) where the distribution is highly peaked, we employ a smaller $\theta_k$ to facilitate the fine-grained local refinement of high-reward modes.

**3. Replica Exchange.** To escape local mode, we periodically propose swapping states $z_t^{(k)} \leftrightarrow z_t^{(k+1)}$ between adjacent chains. The swap is accepted with probability:

$$\alpha_{\text{swap}} = \min\{1,$$
$$\exp\big((\beta_{k+1} - \beta_k)\big[R(T_\theta(z_t^{(k)})) - R(T_\theta(z_t^{(k+1)}))\big]\big)\}.$$

*Rationale:* This mechanism creates a bidirectional flow: high-reward particles found by permissive lower chains are promoted up the ladder ("tunneling" to the target), while particles stuck in low-reward modes in the target chain are demoted to lower temperatures for aggressive re-exploration.

**4. Final Transport.** After $t_{\max}$ iterations, the samples $\{z_{t_{\max}}^{(K)}\}$ from the final chain approximately follow the target distribution $\tilde{q}_\theta$. We obtain the final samples in the data space by pushing these particles using the learned transport map:

$$x_{t_{\max}}^{\text{SPT}} = T_\theta(z_{t_{\max}}^{(K)}). \tag{11}$$

*Remark* 4.1. (**Choice of Source Distribution.**) In this work, we focus on settings where the source distribution $q$ is a standard Gaussian $\mathcal{N}(0, I)$, consistent with the vast majority of FMs and DMs. This choice is critical for algorithmic efficiency in high-dimensional spaces with guarantees for the use of the pCN operator. Extending this framework to non-Gaussian or empirical source distributions (as in Schrödinger Bridge Matching [38]) would require designing complex, geometry-aware proposal kernels to maintain reasonable acceptance rates on arbitrary data manifolds—an open challenge in the Markov chain Monte Carlo (MCMC) literature. We leave this generalization for future work.

### 4.2. Theoretical Analysis of SPT

We derive an explicit error bound for SPT in terms of the number of sampling iterations and the transport map approximation error by decomposing the total error into two components: (i) the approximation error of the learned transport map (Theorem 3.3), and (ii) the sampling error induced by the finite-time mixing of PT. We first establish the convergence properties of PT under our setting in Theorem 4.2, and then combine both error terms in Corollary 4.4 to obtain the final end-to-end bound. The formal statements of Theorem 4.2 and Corollary 4.4, along with their proofs, are provided in Appendix A.

Let $\tilde{q}_t^{\text{SPT}}$ denote the probability density of $z_t^{(K)}$, the samples in the chain at temperature $\beta_K = \beta$ after $t$ iterations. The convergence of the PT algorithm in our setting can be summarized as follows:

**Theorem 4.2** (Informal). *Let $\tilde{q}_t^{\text{SPT}}$ denote the probability density of the random variable $z_t^{(K)}$. Then, there exist a*

*contraction rate $\rho \in (0, 1)$ and a constant $C' > 0$ such that for all $t \geq 0$:*

$$W_2(\tilde{q}_t^{\text{SPT}}, \tilde{q}_\theta) \leq C' \cdot \rho^t. \tag{12}$$

*Remark* 4.3. Our derivation builds upon the $L_2$-spectral gap analysis of PT in [28] and related results for the pCN operator [18; 37]. The convergence rate $\rho$ corresponds to the spectral gap of the PT transition kernel [48; 32] and is governed by the swapping mechanism, which enables the chain to bypass low-probability barriers. Its efficiency, characterized by $\rho$, depends critically on the temperature schedule $\{\beta_k\}$ and the degree of overlap between adjacent distributions $\pi_k$ and $\pi_{k+1}$. In this work, we adopt a standard grid for $\{\beta_k\}$ and leave the empirical study of alternative schedules to future work.

We now establish the total error bound for our SPT framework by combining the sampling convergence with the transport approximation error.

**Corollary 4.4** (Informal). *Let $x_t^{\text{SPT}} = T_\theta(z_t^{(K)})$ denote the output of the SPT algorithm at iteration $t$, and let $\tilde{p}_t^{\text{SPT}} = [T_\theta]_\sharp \tilde{q}_t^{\text{SPT}}$ be its induced probability density. Then the 2-Wasserstein distance between $\tilde{p}_t^{\text{SPT}}$ and the target distribution $\tilde{p}$ satisfies*

$$W_2\left(\tilde{p}, \tilde{p}_t^{\text{SPT}}\right) \leq C \cdot \epsilon + L \cdot C' \rho^t, \tag{13}$$

*where $L > 0$ is a constant, $C, C', \rho$ are the constants appearing in Theorems 3.3 and 4.2, respectively, and $\epsilon = \sqrt{\mathbb{E}_{z \sim q}[\|T(z) - T_\theta(z)\|^2]}$ represents the approximation error of the learned transport map.*

The bound in Corollary 4.4 decouples the total generation error into two distinct sources: map approximation and algorithmic convergence. It explicitly characterizes how the model approximation error $\epsilon$ and the sampling iterations $t$ jointly control the final generation quality. Specifically, as $\epsilon \to 0$ (i.e., the transport map becomes exact) and $t \to \infty$, the upper bound converges to zero, guaranteeing exact recovery of the target tilted distribution.

## 5. Numerical Experiments

In this section, we validate the effectiveness of SPT for test-time steering. We consider a synthetic task and several representative benchmark generative tasks, including conditional image synthesis and dynamical system trajectory sampling. See Appendix B for the full experimental details.

### 5.1. Analytic Examples

To solidify the challenge of existing test-time guidance methods in sampling high-reward but low probability samples, we consider sampling the following one-dimensional (1D)

tilted distribution using state-of-the-art (SOTA) FK Steering (see Appendix B.1 for experimental details):

$$\tilde{p}(x) \propto p(x) \exp(R(x)), \tag{14}$$

where

$$p(x) = 0.9\mathcal{N}(x| - 1.5, (.02)^2) + 0.1\mathcal{N}(x|1.5, (.02)^2),$$

$$R(x) = 1.5e^{-\frac{(x+1.5)^2}{3}} + 2e^{-\frac{(x-1.5)^2}{\omega^2}}.$$

The majority of the mass of $p(x)$ is supplied by $0.9\mathcal{N}(x| - 1.5, (.02)^2)$, while $0.1\mathcal{N}(x|1.5, (.02)^2)$ contains little mass. However, $R(x)$ assigns a larger reward to the low-mass (right) mode, making it the global optimum of the reward tilted distribution, while the high-mass (left) mode becomes a local optimum. This setup makes it crucial to effectively generate low-probability samples to achieve a high reward. Moreover, the parameter $\omega^2$ in the reward function $R(x)$ controls the sharpness of $\tilde{p}(x)$—smaller values of $\omega^2$ produce a more sharply peaked distribution near 1.5.

In contrast to FK Steering (see Fig. 1(a)), SPT can effectively cover the high-reward mode of $\tilde{p}(x)$ at $x = 1.5$ for $\omega^2 = 0.01$ (see Fig. 1(b)). Quantitatively, SPT outperforms FK Steering across different $\omega^2$ values in terms of Wasserstein distance to the true reward-adjusted distribution. As shown in Figure 1(c), SPT maintains low Wasserstein error as a function of $\omega^2$, whereas FK Steering degrades rapidly as $\omega^2$ decreases, highlighting its limited ability to locate narrow, high-reward modes. In contrast, SPT maintains a very small error for different $\omega^2$.

The experimental setup is provided in Appendix B.1.1. We extend the analysis to a 2D four-component Gaussian mixture to assess robustness under higher dimensionality and more complex multimodal structure, with full model, training, and sampling configurations available in Appendix B.1.3. Additionally, we perform an ablation over guidance strength $\beta$ to study its effect on the trade-off between exploration and exploitation in Appendix B.1.2.

## 5.2. Text to Image Guidance

We compare SPT against FK Steering and other SOTA training-based alignment methods for DMs, including direct preference optimization (DPO) and diffusion-DPO (DDPO) [46; 6], on the task of steering pretrained text-to-image DMs toward more aesthetically pleasing samples. We follow the ImageReward benchmark [49] for aesthetic alignment, testing on their benchmark prompts and steering with their Human Preference Model. We find that SPT consistently produces samples with higher aesthetic preference alignment than FK Steering, the unsteered Stable Diffusion baseline [36], and fine-tuned DPO/DDPO models. In addition, when comparing the diversity of generated samples, SPT outperforms FK Steering and the base Stable Diffusion

models, highlighting its ability to improve both preference alignment and diversity of generated samples.

Following the experimental design used in [39], Table 1 reports the average ImageReward score of the highest-quality sample generated per prompt. SPT consistently achieves the highest generation quality across all model variants. Most notably, SPT achieves a 57% relative improvement over FK Steering on Stable Diffusion v1.5 (1.411 vs 0.898), and delivers substantial gains on Stable Diffusion 1.4 (52% improvement) and Stable Diffusion XL (14% improvement). These results suggest that SPT's sampling mechanism more effectively explores the reward landscape to identify high-quality generations compared to existing steering approaches.

| Model | Method | ImageReward ↑ |
|---|---|---|
| Stable Diffusion 1.4 | Base [36] | 0.234 |
| | DDPO [6] | 0.263 |
| | Best-of-K [46] | 0.800 |
| | FK Steering [39] | 0.927 |
| | SPT (**ours**) | **1.409** |
| Stable Diffusion 1.5 | Base [36] | 0.187 |
| | DPO[46] | 0.343 |
| | Best-of-K [46] | 0.737 |
| | FK Steering [39] | 0.898 |
| | SPT (**ours**) | **1.411** |
| Stable Diffusion XL | Base [36] | 0.871 |
| | DPO [46] | 0.859 |
| | Best-of-K [46] | 1.236 |
| | FK Steering | 1.298 |
| | SPT (**ours**) | **1.479** |

*Table 1.* Comparison of the highest quality generated sample ImageReward between SPT, FK Steering, and various baselines. Base, DDPO, DPO, and FK Steering results are from [39].

Under the same experimental setting, we further compare the diversity of samples generated by SPT, FK Steering, and the unsteered (base) model using a CLIP-based diversity metric [35]. Following [39], we define CLIP diversity as the average pairwise squared $\ell_2$ distance between CLIP image embeddings (see details in Appendix B.3). Table 2 reports CLIP diversity scores for samples generated by FK Steering, an unsteered Stable Diffusion baseline, and SPT. Across all evaluated Stable Diffusion variants, FK Steering exhibits the lowest diversity. This is consistent with its resampling scheme, which tends to collapse toward a narrow set of high-reward samples. While the unsteered baseline maintains a natural variety due to unconstrained sampling, it lacks a mechanism to actively navigate the semantic landscape. Notably, SPT achieves significantly higher diversity than both FK Steering and the base model. This suggests that SPT allows exploration over disjoint modes of the distribution, allowing for the discovery of a more diverse set of

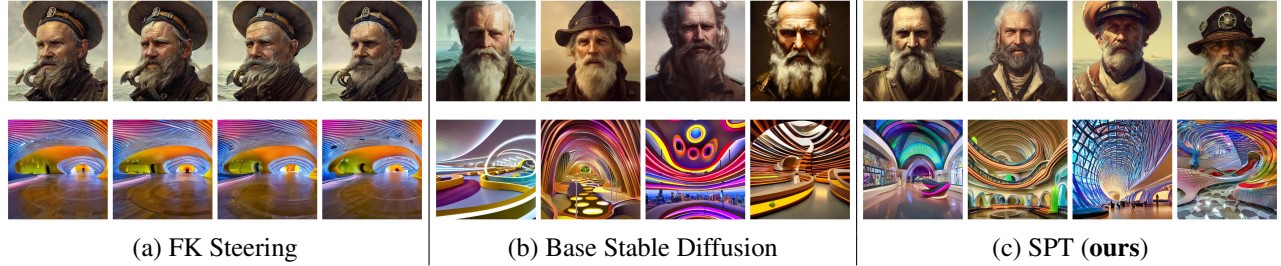

|  (a) FK Steering | (b) Base Stable Diffusion | (c) SPT (**ours**) |

*Figure 2.* Comparison of samples generated with Stable Diffusion v1.4 using three methods: Base Stable Diffusion (no steering), FK Steering, and SPT. Prompt (first row): *Portrait of an old sea captain, male, detailed face, fantasy, highly detailed, cinematic, art painting by greg rutkowski.* Prompt (second row): *extremely detailed stunning beautiful futuristic smooth curvilinear museum interior, colorful, hyper, real.*

high-quality samples that standard sampling fails to reach.

| Model | Method | CLIP Div ↑ |
|---|---|---|
| | FK Steering [39] | 0.193 |
| Stable Diffusion 1.4 | Base [36] | 0.316 |
| | SPT (**ours**) | **0.346** |
| | FK Steering [39] | 0.123 |
| Stable Diffusion XL | Base [36] | 0.286 |
| | SPT (**ours**) | **0.351** |

*Table 2.* Comparison of the generated sample diversity between the base Stable Diffusion model (no steering), FK Steering, and SPT. For FK Steering, we report the strongest results from [39] for Stable Diffusion 1.4 and Stable Diffusion XL; results for Stable Diffusion 1.5 were not available in their work.

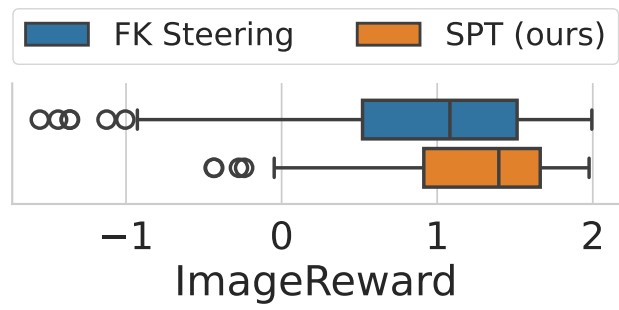

*Figure 3.* Efficiency comparison on the ImageReward benchmark. We report the distribution of ImageReward scores for FK Steering and SPT under a fixed budget of 5,250 evaluations. SPT demonstrates a statistically significant improvement over the baseline, with a Welch's t-test p-value of $1.3 \times 10^{-11}$.

Additionally, we demonstrate the superior efficiency of SPT compared to FK Steering by evaluating the generation quality under a strictly controlled budget of 5,250 vector field evaluations (NFE). Matching the NFE allows us to compare each method's ability to optimize reward given equal computational budget. As shown in Fig. 3, SPT achieves a substantially higher median ImageReward than FK Steering across the ImageReward benchmark [49]. To rigorously confirm this improvement, we conducted a Welch's one-sided two-sample t-test (unequal variances) on 400 generated images per method ($\alpha = 0.05$). The analysis yields a p-value of $1.3 \times 10^{-11}$, providing overwhelming statistical evidence that SPT significantly outperforms FK Steering in generating high-reward samples. Full experimental hyper-parameters are detailed in Appendix B.3.1.

### 5.3. Sampling dynamical system trajectories

We consider the task of steering a GM to sample dynamical system trajectories that are representative of a user-defined event $E$ [14]. Trajectories are a discrete time series of $M$ dynamical system states of dimension $d$ concatenated such that $\boldsymbol{x} = [\boldsymbol{x}(\tau_m)]_{m=1}^M \in \mathbb{R}^{Md}$. The trajectories of $E$ conform to some constraint (e.g., $C(\boldsymbol{x}) > 0$). We

want to steer a GM to sample $\boldsymbol{x} \sim p(\boldsymbol{x}|E)$. We assume $p(\boldsymbol{x}|E) \propto p(\boldsymbol{x})p(E|\boldsymbol{x}) = p(\boldsymbol{x})\exp(\beta R(\boldsymbol{x}))$. Inclusion in $E$ is a binary condition, so we choose a *discontinuous* $R$; specifically, $R(\boldsymbol{x}) = \operatorname{sign}(C(\boldsymbol{x}))$.

Following [14], we consider two dynamical systems and events—Lorenz '63 [31], whose event is when $\boldsymbol{x}$ is on one arm of the chaotic attractor, and FitzHugh-Nagumo [15; 33], whose event is neuron spiking. Precise characterizations are given in Appendix B.2. The GM is a flow matching model [29] with a variance-exploding probability path [43] with Euler-Maruyama used for FK Steering and Heun for SPT.

We compare FK Steering and SPT by their steering success rate, the fraction of trajectories $\boldsymbol{x}$ generated such that $\boldsymbol{x} \in E$. The parameters of FK Steering and SPT are found using grid search to maximize success rate (see Appendix B.2), and we report the best results in Table 3. Lorenz '63 and FitzHugh-Nagumo have an empirical event probability of $p(E) \approx 0.20$ and $p(E) \approx 1/30$, respectively [14; 23]. SPT increases the odds of generating $\boldsymbol{x} \in E$ to near 100% for Lorenz '63 and to 81% for FitzHugh-Nagumo. In contrast, while FK Steering increases the odds to 86% for Lorenz '63, it is less effective for FitzHugh-Nagumo, only increasing the odds to 27%. SPT more successfully steers compared to FK

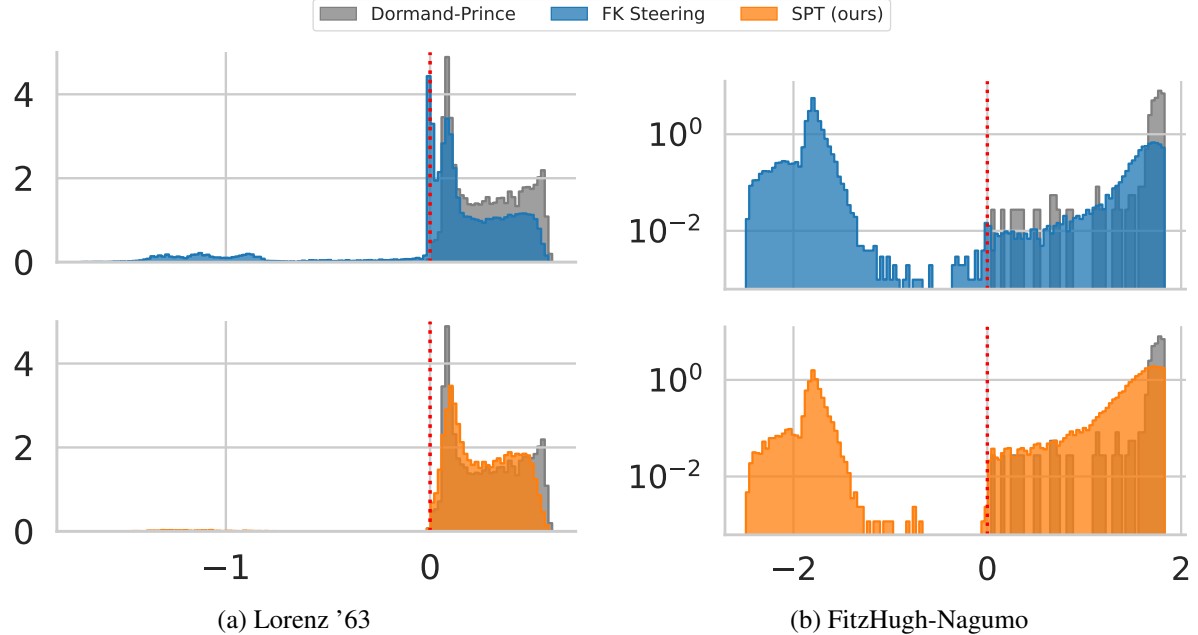

*Figure 4.* Distribution of $C(\boldsymbol{x})$ where $\boldsymbol{x}$ is sampled using Dormand-Prince [11], FK Steering, or SPT. Parameters used for FK Steering and SPT are in Table 3.

Steering with this discontinuous reward function, especially when the events have low probability.

| Lorenz '63 ($\beta = 8.29$) | | |
|---|---|---|
| **Method** | **SR ($\uparrow$)** | **TV ($\downarrow$)** |
| Best-of-$N$ | 0.19 | 0.81 |
| FK Steering($S = 8, \mathrm{P} = \mathrm{Diff.}$) | 0.86 | 0.30 |
| SPT($K = 4, t_{\max} = 8$) **(ours)** | **0.99** | **0.14** |
| FitzHugh-Nagumo ($\beta = 10.27$) | | |
| Best-of-$N$ | 0.04 | 0.98 |
| FK Steering($S = 8, \mathrm{P} = \mathrm{Diff.}$) | 0.27 | 0.86 |
| SPT($K = 8, t_{\max} = 8$) **(ours)** | **0.81** | **0.64** |

*Table 3.* The success rate (SR) of SPT and FK (with ensemble size $S$ and potential function P), and the total variation (TV) distance between the distributions of Fig. 4.

Steering samples should follow $p(\boldsymbol{x}|E)$; otherwise, sampling the same event trajectory gives a 100% success rate. Figure 4 shows the distributions of $C(\boldsymbol{x})$ from $\boldsymbol{x}$ sampled using FK Steering or SPT, with the total variation (TV) distance given in Table 3. SPT more closely approximates $p(\boldsymbol{x}|E)$ for both systems.

## 6. Concluding Remarks

We have introduced *Source Parallel Tempering (SPT)*, a gradient-free test-time guidance framework that effectively steers flow-based models by performing parallel tempering in the source space. By coupling dimension-independent local exploration with global replica exchange, SPT effectively navigates multimodal and non-differentiable reward landscapes. Moreover, we provide an end-to-end theoretical analysis that rigorously links the model's $L_2$ training error to test-time guided generation quality.

Looking ahead, several avenues remain for investigation. While SPT currently uses a gradient-free exploration kernel, incorporating gradient-informed strategies (e.g., Langevin dynamics) or adaptive MCMC could further accelerate convergence when differentiable rewards are available. Finally, extending source-space tempering to discrete domains—such as discrete diffusion models or large language models—presents a promising path for controllable text generation without reinforcement learning.

## Acknowledgement

This material is based on research sponsored by NSF grants DMS-2152762, DMS-2208361, DMS-2219956, DMS-2208356, and DMS-2436344, and DOE grants DE-SC0023490, DE-SC0025589, and DE-SC0025801. This work is also supported by NIH grants R01HL16351 and P41EB032840

## Impact Statement

This paper presents work whose goal is to advance the field of machine learning. There are many potential societal consequences of our work, none of which we feel must be specifically highlighted here.

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

# A. Proofs and Theoretical Analysis.

In this section, we provide the detailed proofs for the theoretical results presented in the main text. We begin by establishing the necessary mathematical preliminaries and assumptions.

## A.1. Preliminaries and Assumptions

We first state the standard definitions for the functional inequalities used in our analysis.

**Definition A.1** (Log-Sobolev Inequality). We say a probability density $\mu$ satisfies the Log-Sobolev Inequality with constant $C_{\mathrm{LSI}} > 0$ if for all densities $\rho$ absolutely continuous w.r.t $\mu$:

$$\mathrm{KL}(\rho\|\mu) \leq \frac{1}{2C_{\mathrm{LSI}}}\mathcal{I}(\rho\|\mu), \tag{15}$$

where $\mathrm{KL}(\rho\|\mu) = \int \log \frac{\rho(\boldsymbol{z})}{\mu(\boldsymbol{z})}\rho(\boldsymbol{z})d\boldsymbol{z}$ is the Kullback-Leibler divergence and $\mathcal{I}(\rho\|\mu) := \int \|\nabla \log \frac{\rho(\boldsymbol{z})}{\mu(\boldsymbol{z})}\|^2\rho(\boldsymbol{z})d\boldsymbol{z}$ is the relative Fisher Information.

**Definition A.2** (Talagrand's Inequality ($T_2$)). We say a probability measure $\mu$ satisfies Talagrand's transport inequality $T_2$ with constant $C_{T_2} > 0$ if for all densities $\rho$ absolutely continuous w.r.t $\mu$:

$$W_2^2(\rho, \mu) \leq \frac{2}{C_{T_2}} \mathrm{KL}(\rho\|\mu), \tag{16}$$

**Definition A.3** (Poincaré Inequality). We say a probability measure $\mu$ satisfies the Poincaré inequality with constant $C_P > 0$ if for all bounded smooth functions $f : \mathbb{R}^d \to \mathbb{R}$:

$$\mathrm{Var}_\mu(f) \leq C_P \mathbb{E}_\mu\left[\|\nabla f\|^2\right], \tag{17}$$

where $\mathrm{Var}_\mu(f) := \mathbb{E}_\mu[f^2] - (\mathbb{E}_\mu[f])^2$ is the variance of $f$ under $\mu$.

Next, we restate our core assumptions regarding the target geometry and the learned map.

**Assumption A.4.** The measure of the source probability density $q$ is absolutely continuous with respect to the Lebesgue measure and $q$ satisfies the Poincaré inequality with constant $C_q$.

*Remark* A.5. Assumption A.4 is naturally satisfied by the standard Gaussian prior $q = \mathcal{N}(\boldsymbol{0}, \mathbf{I})$ used in continuous GMs (e.g., DM, FM). The Gaussian measure satisfies the Poincaré inequality with a constant $C_q = 1$ [3].

**Assumption A.6.** The learned transport map $T_\theta$ is $L_T$-Lipschitz. The reward function $R$ is $L_R$-Lipschitz and bounded by a constant $M_R$ (i.e., $|R(\boldsymbol{z})| \leq M_R$ for all $\boldsymbol{z}$).

The assumption that the transport map $T_\theta$ is Lipschitz continuous is a direct consequence of standard regularity conditions imposed on the learned vector fields or score functions. In the theoretical analysis of FM [29; 2; 30] and DMs [10], it is standard to assume that the learned vector field $v_\theta(t, \boldsymbol{x})$ or score $s_\theta(t, \boldsymbol{x})$ is Lipschitz continuous. By the classical theory of Ordinary Differential Equations (specifically Grönwall's inequality), a Lipschitz drift implies that the induced flow map $T_\theta$ is also Lipschitz continuous, with a constant depending exponentially on the time horizon [47].

The Lipschitz continuity of $R$ is a necessary condition to control error propagation; specifically, it ensures that the approximation error between the transport maps, $\|T(\boldsymbol{z}) - T_\theta(\boldsymbol{z})\|$, does not result in an unbounded divergence between the reward values $R(T(\boldsymbol{z}))$ and $R(T_\theta(\boldsymbol{z}))$. Furthermore, the boundedness assumption is mild and typically satisfied in practice, as most utility functions operate within a finite range (e.g., ImageReward scores fall within $[-2, 2]$).

## A.2. Formal version of Theorem 3.3:

We are now ready to state the formal version of Theorem 3.3:

**Theorem A.7.** *Let*

$$\epsilon^2 = \mathbb{E}_{\boldsymbol{z}\sim q}[\|T(\boldsymbol{z}) - T_\theta(\boldsymbol{z})\|^2]$$

*be the expected squared error of the learned map, which is upper bounded by the training loss used to fit $T_\theta$ (see Sec. 2). Under Assumptions A.4 and A.6, the 2-Wasserstein distance between the tilted target distribution $\tilde{p}$ and its approximation $[T_\theta]_\sharp \tilde{q}_\theta$ satisfies:*

$$W_2\left(\tilde{p}, [T_\theta]_\sharp \tilde{q}_\theta\right) \leq (1 + \beta L_T L_R C_q \cdot e^{2\beta M_R})e^{\beta M_R} \cdot \epsilon., \tag{18}$$

*where $C_q$ is the Poincaré constant of the prior $q$.*

Before proving Theorem A.7, we establish a lemma bounding the importance weights along an interpolation path between $\tilde{q}$ and $\tilde{q}_\theta$:

**Lemma A.8.** *Let* $\rho_t(\boldsymbol{z}) = \frac{1}{Z_t} q(\boldsymbol{z}) \exp(V_t(\boldsymbol{z}))$ *where* $V_t(\boldsymbol{z}) = (1 - t)\beta R(T(\boldsymbol{z})) + t\beta R(T_\theta(\boldsymbol{z}))$ *and* $Z_t = \int q(\boldsymbol{z}) \exp(V_t(\boldsymbol{z}))$. *If* $|R(\boldsymbol{z})| \leq M_R$, *then the importance weights* $w_t(\boldsymbol{z}) = \frac{\rho_t(\boldsymbol{z})}{q(\boldsymbol{z})}$ *are bounded by* $e^{2\beta M_R}$ *for all* $t \in [0, 1]$.

*Proof of Lemma A.8.* Since $|R| \leq M_R$, the potential $V_t$ is uniformly bounded:

$$-\beta M_R \leq V_t(\boldsymbol{z}) \leq \beta M_R.$$

First, we bound the normalization constant $Z_t = \mathbb{E}_{\boldsymbol{z} \sim q}[\exp(V_t(\boldsymbol{z}))]$:

$$\exp(-\beta M_R) \leq Z_t \leq \exp(\beta M_R).$$

Now we bound the ratio:

$$\frac{\rho_t(\boldsymbol{z})}{q(\boldsymbol{z})} = \frac{\exp(V_t(\boldsymbol{z}))}{Z_t} \leq \frac{\exp(\beta M_R)}{\exp(-\beta M_R)} = \exp(2\beta M_R).$$

$\square$

*Proof of Theorem A.7.* By Proposition 3.1, $\tilde{p} = [T]_\sharp \tilde{q}$, and the triangle inequality, we decompose the error as

$$W_2(\tilde{p}, [T_\theta]_\sharp \tilde{q}) = W_2([T]_\sharp \tilde{q}, [T_\theta]_\sharp \tilde{q}) \leq W_2([T]_\sharp \tilde{q}, [T_\theta]_\sharp \tilde{q}) + W_2([T_\theta]_\sharp \tilde{q}, [T_\theta]_\sharp \tilde{q}_\theta) \tag{19}$$

By the definition of Wasserstein distance, we have

$$
\begin{aligned}
W_2^2([T]_\sharp \tilde{q}, [T_\theta]_\sharp \tilde{q}) &\leq \mathbb{E}_{\boldsymbol{z} \sim \tilde{q}}\left[\|T(\boldsymbol{z}) - T_\theta(\boldsymbol{z})\|^2\right] \\
&= \mathbb{E}_{\boldsymbol{z} \sim q}\left[\frac{\tilde{q}(\boldsymbol{z})}{q(\boldsymbol{z})}\|T(\boldsymbol{z}) - T_\theta(\boldsymbol{z})\|^2\right] \\
&\leq \left(\sup_{\boldsymbol{z} \in \mathrm{supp}(q)} \frac{\tilde{q}(\boldsymbol{z})}{q(\boldsymbol{z})}\right) \cdot \mathbb{E}_{\boldsymbol{z} \sim q}\left[\|T(\boldsymbol{z}) - T_\theta(\boldsymbol{z})\|^2\right] \text{ (Apply Hölder's inequality)} \\
&\leq e^{2\beta M_R} \cdot \epsilon^2. \text{ (Apply Lemma A.8)}
\end{aligned}
\tag{20}
$$

On the other hand, we first notice that,

$$W_2([T_\theta]_\sharp \tilde{q}, [T_\theta]_\sharp \tilde{q}_\theta) \leq L_T \cdot W_2(\tilde{q}, \tilde{q}_\theta). \tag{21}$$

To bound $W_2(\tilde{q}, \tilde{q}_\theta)$, we construct a path between $\tilde{q}$ and $\tilde{q}_\theta$. Let $V_t(\boldsymbol{z}) = (1 - t)\beta R(T(\boldsymbol{z})) + t\beta R(T_\theta(\boldsymbol{z}))$. We define the path of measures:

$$\rho_t(\boldsymbol{z}) = \frac{1}{Z_t} q(\boldsymbol{z}) \exp(V_t(\boldsymbol{z})), \quad t \in [0, 1], \tag{22}$$

such that $\rho_0 = \tilde{q}$ and $\rho_1 = \tilde{q}_\theta$ where $Z_t = \int q(\boldsymbol{z}) \exp(V_t(\boldsymbol{z}))$.

Under Assumption A.4, we can apply the Benamou-Brenier formula:

$$W_2(\tilde{q}, \tilde{q}_\theta) \leq \int_0^1 \|\boldsymbol{v}_t\|_{L^2(\rho_t)} dt, \tag{23}$$

where $\boldsymbol{v}_t$ is a vector field satisfying the continuity equation $\partial_t \rho_t + \nabla \cdot (\rho_t v_t) = 0$.

Since $q$ satisfies a Poincaré inequality with constant $C_q$ and $|R(\boldsymbol{z})| \leq M_R$ for all $\boldsymbol{z}$, the Holley-Stroock perturbation lemma [22] ensures that $\rho_t$ also satisfies a Poincaré inequality with a constant $C_{\rho_t} \leq C_q \cdot e^{2\beta M_R}$.

Consider the solution $\psi_t$ to the weighted Poisson equation:

$$\nabla \cdot (\rho_t \nabla \psi_t) = -\partial_t \rho_t = -\rho_t(\partial_t V_t - \mathbb{E}_{\rho_t}[\partial_t V_t]). \tag{24}$$

Without loss of generality, we can assume $\mathbb{E}_{\rho_t}[\psi_t] = 0$ (by considering the solution $\psi_t - \mathbb{E}_{\rho_t}[\psi_t]$). Multiplying by $\psi_t$ and integrating by parts yields the identity:

$$\|\boldsymbol{v}_t\|_{L^2(\rho_t)}^2 = \int \nabla \psi_t \cdot \rho_t \nabla \psi_t = \int \psi_t(-\nabla \cdot (\rho_t \nabla \psi_t)) = \int \psi_t \rho_t(\partial_t V_t) \leq \|\psi_t\|_{L^2(\rho_t)} \cdot \|\partial_t V_t\|_{L^2(\rho_t)}. \tag{25}$$

The Poincaré inequality implies that:

$$\|\psi_t\|_{L^2(\rho_t)} = \|\psi_t - \mathbb{E}[\psi_t]\|_{L^2(\rho_t)} \leq C_{\rho_t} \cdot \|\nabla \psi_t\|_{L^2(\rho_t)} = C_{\rho_t} \cdot \|\boldsymbol{v}_t\|_{L^2(\rho_t)} \tag{26}$$

We then have:

$$\|\boldsymbol{v}_t\|_{L^2(\rho_t)} \leq C_q \cdot e^{2\beta M_R} \|\partial_t V_t\|_{L^2(\rho_t)}. \tag{27}$$

The time derivative of the potential is:

$$\partial_t V_t(\boldsymbol{z}) = \beta R(T_\theta(\boldsymbol{z})) - \beta R(T(\boldsymbol{z})). \tag{28}$$

Using the Lipschitz property of $R$, we have the pointwise bound $|\dot{V}_t(\boldsymbol{z})| \leq \beta L_R |T_\theta(\boldsymbol{z}) - T(\boldsymbol{z})|$. Substituting this into the integral:

$$\begin{aligned} W_2(\tilde{q}, \tilde{q}_\theta) &\leq C_q \cdot e^{2\beta M_R} \beta L_R \int_0^1 |T(\boldsymbol{z}) - T_\theta(\boldsymbol{z})|_{L^2(\rho_t)} dt \\ &\leq \beta L_R C_q \cdot e^{2\beta M_R} \sup_{t \in [0,1]} \sqrt{\mathbb{E}_{\boldsymbol{z} \sim \rho_t}[|T(\boldsymbol{z}) - T_\theta(\boldsymbol{z})|^2]}. \end{aligned} \tag{29}$$

Finally, according to Lemma A.8, the importance weights $\frac{\rho_t(\boldsymbol{z})}{q(\boldsymbol{z})}$ are uniformly bounded by $e^{2\beta M_R}$. This allows us to relate the error under $\rho_t$ to the error under $q$:

$$\sup_{t \in [0,1]} \sqrt{\mathbb{E}_{\boldsymbol{z} \sim \rho_t}[|T(\boldsymbol{z}) - T_\theta(\boldsymbol{z})|^2]} \leq e^{\beta M_R} \sqrt{\mathbb{E}_{\boldsymbol{z} \sim q}[|T(\boldsymbol{z}) - T_\theta(\boldsymbol{z})|^2]} \leq e^{\beta M_R} \epsilon. \tag{30}$$

Substituting this back into the earlier bound:

$$W_2(\tilde{q}, \tilde{q}_\theta) \leq \beta L_R C_q e^{3\beta M_R} \epsilon. \tag{31}$$

The total bound becomes:

$$W_2\left(\tilde{p}, [T_\theta]_\sharp \tilde{q}_\theta\right) \leq (e^{\beta M_R} + \beta L_T L_R C_q \cdot e^{2\beta M_R} e^{\beta M_R})\epsilon = (1 + \beta L_T L_R C_q \cdot e^{2\beta M_R})e^{\beta M_R} \cdot \epsilon. \tag{32}$$

This completes the proof.

$\square$

## A.3. Formal version of Theorem 4.2 and Corollary 4.4

Throughout this subsection, we assume that $q \sim \mathcal{N}(\mathbf{0}, \boldsymbol{I})$, consistent with the assumption adopted in the formulation of our framework. We now state the formal version of Theorem 4.2.

**Theorem A.9.** *Under Assumption A.6, let $\tilde{q}_t^{\text{SPT}}$ denote the probability density of the random variable $\boldsymbol{z}_t^{(K)}$. Then, there exists a contraction rate $\rho \in (0, 1)$ such that for all $t \geq 0$:*

$$W_2(\tilde{q}_t^{\text{SPT}}, \tilde{q}_\theta) \leq \sqrt{2}e^{2\beta M_R} \cdot \rho^t. \tag{33}$$

To prove Theorem A.9, we first establish the link between the Wasserstein distance and the $\chi^2$-divergence.

**Lemma A.10.** *If the reward function is bounded such that $|R(\cdot)| \leq M_R$, then for any probability density $\rho$ absolutely continuous w.r.t $\tilde{q}_\theta$, we have*

$$W_2^2(\rho, \tilde{q}_\theta) \leq 2e^{2\beta M_R} \chi^2(\rho \| \tilde{q}_\theta), \tag{34}$$

*where $\chi^2(\rho \| \mu) = \int \left(\frac{\rho(\boldsymbol{z})}{\mu(\boldsymbol{z})} - 1\right)^2 \mu(\boldsymbol{z}) d\boldsymbol{z}$ is the chi-squared divergence.*

*Proof of Lemma A.10.* First, a classical result by Gross [17] (see also [1]) establishes that the standard Gaussian distribution $q \sim \mathcal{N}(\mathbf{0}, \boldsymbol{I})$ satisfies the LSI with constant $C_{\mathrm{LSI}}(q) = 1$.

Due to the boundedness of the reward function $R$, we invoke the perturbation lemma by Holley & Stroock [22]. This result ensures that since $\tilde{q}_\theta$ is a bounded perturbation of $q$, it retains the LSI property. Specifically, the new LSI constant satisfies:

$$C_{\mathrm{LSI}}(\tilde{q}_\theta) = C_{\mathrm{LSI}}(q) \cdot e^{-2\beta M_R} = e^{-2\beta M_R}.$$

Next, by the result in [34], we derive that $\tilde{q}_\theta$ satisfies Talagrand's inequality $T_2$ with constant $C_{T_2}$ determined by its LSI constant. Specifically, since LSI implies $T_2$ with $C_{T_2} = C_{\mathrm{LSI}}$, we have for any probability density $\rho$, the 2-Wasserstein distance is bounded by the KL divergence:

$$W_2^2(\rho, \tilde{q}_\theta) \leq 2e^{2\beta M_R} \mathrm{KL}(\rho \| \tilde{q}_\theta). \tag{35}$$

Finally, we use the standard inequality $\mathrm{KL}(\rho \| \nu) \leq \chi^2(\rho \| \nu)$ to loosen the bound to the $\chi^2$-divergence:

$$W_2^2(\rho, \tilde{q}_\theta) \leq 2e^{2\beta M_R} \chi^2(\rho \| \tilde{q}_\theta).$$

$\square$

*Proof of Theorem A.9.* First, we observe that the preconditioned Crank-Nicolson (pCN) proposal admits an $L_2$ spectral gap [18]. Consequently, as established by [28], the full parallel tempering algorithm inherits this property, possessing a strictly positive spectral gap $\lambda > 0$. This implies geometric ergodicity in the $\chi^2$-divergence:

$$\chi^2(\rho_t \| \tilde{q}_\theta) \leq e^{-2\lambda t} \chi^2(\rho_0 \| \tilde{q}_\theta) = e^{-2\lambda t} \chi^2(q \| \tilde{q}_\theta). \tag{36}$$

Combining the spectral decay in equation 36 with the transport-variance bound from Lemma A.10, we obtain:

$$W_2^2(\rho_t, \tilde{q}_\theta) \leq 2e^{2\beta M_R} \left( e^{-2\lambda t} \chi^2(q \| \tilde{q}_\theta) \right). \tag{37}$$

Furthermore, using the boundedness of $R$, we have $\chi^2(q \| \tilde{q}_\theta) \leq e^{2\beta M_R} - 1 < e^{2\beta M_R}$. Substituting this and taking the square root of both sides in equation 37 yields the final bound:

$$W_2(\rho_t, \tilde{q}_\theta) < \sqrt{2} e^{2\beta M_R} \cdot \underbrace{e^{-\lambda t}}_{\rho^t}.$$

Defining the contraction rate $\rho = e^{-\lambda} \in (0, 1)$, we recover the stated result:

$$W_2(\rho_t, \tilde{q}_\theta) \leq \sqrt{2} e^{2\beta M_R} \cdot \rho^t.$$

$\square$

**Corollary A.11** (Formal Version of Corollary 4.4)**.** *Let $\boldsymbol{x}_t^{\mathrm{SPT}} = T_\theta(\boldsymbol{z}_t^{(K)})$ denote the output of the SPT algorithm at iteration $t$, and let $\tilde{p}_t^{\mathrm{SPT}}$ be its induced probability density. Under Assumptions A.6 and q is Gaussian, the 2-Wasserstein distance to the target is bounded by:*

$$W_2\left(\tilde{p}, \tilde{p}_t^{\mathrm{SPT}}\right) \leq (1 + \beta L_T L_R \cdot e^{2\beta M_R}) e^{\beta M_R} \cdot \epsilon + L_T \cdot \sqrt{2} e^{2\beta M_R} \cdot \rho^t, \tag{38}$$

*where $\epsilon$ represents the approximation error of the learned transport map.*

*Proof of Corollary A.11.* We apply the triangle inequality to bound the total error via the intermediate distribution $[T_\theta]_\sharp \tilde{q}_\theta$:

$$W_2\left(\tilde{p}, \tilde{p}_t^{\mathrm{SPT}}\right) \leq W_2\left(\tilde{p}, [T_\theta]_\sharp \tilde{q}_\theta\right) + W_2\left([T_\theta]_\sharp \tilde{q}_\theta, \tilde{p}_t^{\mathrm{SPT}}\right). \tag{39}$$

From Theorem A.7, we have:

$$W_2\left(\tilde{p}, [T_\theta]_\sharp \tilde{q}_\theta\right) \leq (1 + \beta L_T L_R \cdot e^{2\beta M_R}) e^{\beta M_R} \cdot \epsilon, \tag{40}$$

where we use $C_q = 1$ for a Gaussian distribution.

Note that $\tilde{p}_t^{\mathrm{SPT}} = [T_\theta]_\sharp \tilde{q}_\theta^{\mathrm{SPT}}$ and the intermediate target is $[T_\theta]_\sharp \tilde{q}_\theta$. Since the map $T_\theta$ is $L_T$-Lipschitz, we can bound the distance in the data space by the distance in the source space:

$$W_2\left([T_\theta]_\sharp \tilde{q}_\theta, [T_\theta]_\sharp \tilde{q}_t^{\mathrm{SPT}}\right) \le L_T \cdot W_2\left(\tilde{q}_\theta, \tilde{q}_t^{\mathrm{SPT}}\right). \tag{41}$$

Using the geometric convergence result from Theorem A.9:

$$W_2\left(\tilde{q}_\theta, \tilde{q}_t^{\mathrm{SPT}}\right) \le \sqrt{2}e^{2\beta M_R}\rho^t. \tag{42}$$

Combining these yields the final bound. $\qquad\square$

### A.4. Duality Result and Proof

In this subsection, we provide a concise proof of Proposition 3.1 from [47].

**Proposition 3.1** (Wang et al. [47, Theorem 1])**.** *The pushforward of the tilted source distribution $\tilde{q}$ under the map $T$ is exactly the tilted target distribution $\tilde{p}$. That is, $\tilde{p} = [T]_\sharp \tilde{q}$.*

*Proof of Proposition 3.1.* By equation 5, we have

$$\begin{aligned}
\tilde{p}(\boldsymbol{x}) &= \frac{1}{Z_1} q(\boldsymbol{z}) \left|\det \nabla_{\boldsymbol{z}} T(\boldsymbol{z})\right|^{-1} e^{\beta R(T(\boldsymbol{z}))} \\
&= \frac{1}{Z_1} \tilde{q}(\boldsymbol{z}) \left|\det \nabla_{\boldsymbol{z}} T(\boldsymbol{z})\right|^{-1} \\
&= [T]_\sharp \tilde{q}(\boldsymbol{z}).
\end{aligned} \tag{43}$$

$\qquad\square$

## B. Experimental Details

### B.1. Toy Example: Experimental Setup

#### B.1.1. PRIMARY TOY SETTING: ONE-DIMENSIONAL MIXTURE

**Experimental Settings.** We evaluate the behavior of our sampling methods on a simple 1D Gaussian mixture, focusing on how they handle sharp and low-probability minima. The following hyperparameters define the training and sampling procedures used in Section 5.1.

**Data Distribution.** We train a score model on a two-component Gaussian mixture in $\mathbb{R}$:

$$p(x) = 0.9\,\mathcal{N}(-1.5, 0.02^2) + 0.1\,\mathcal{N}(1.5, 0.02^2), \tag{44}$$

forming an imbalanced bimodal distribution with a dominant local mode and a low-mass secondary mode.

**Diffusion Formulation.** We use a variance-exploding SDE with noise scale $\sigma = 4.0$, forward dynamics

$$dx = g(t)\,dW_t, \quad g(t) = \sigma^t, \tag{45}$$

and the perturbation kernel

$$x_t = x + \sigma(t)z, \quad z \sim \mathcal{N}(0, I), \quad \sigma(t) = \sqrt{\frac{\sigma^{2t} - 1}{2\log\sigma}}. \tag{46}$$

The score network $s_\theta(x, t)$ is trained using denoising score matching:

$$\mathcal{L}(\theta) = \mathbb{E}_{x,t,z}\left[\left\|s_\theta(x_t, t)\sigma(t) + z\right\|_2^2\right]. \tag{47}$$

**Sampling Dynamics.**

- **Stochastic reverse SDE (FK Steering):** $dx = -g(t)^2 \nabla_x \log p_t(x)\, dt + g(t)\, d\bar{W}_t$

- **Deterministic probability flow ODE (SPT):** $dx = -\frac{1}{2}g(t)^2 \nabla_x \log p_t(x)\, dt$

Both dynamics share identical marginal distributions $p_t(x)$; the SDE produces stochastic trajectories, while the ODE defines a deterministic transport map.

**Reward-Guided Distribution.** Sampling targets the tilted distribution

$$p_\beta(x) \propto p(x) \exp(\beta R(x)), \quad R(x) = 1.5\, e^{-\frac{(x+1.5)^2}{3}} + 2\, e^{-\frac{(x-1.5)^2}{\omega^2}}, \tag{48}$$

where $\omega^2$ controls the sharpness of the global reward minimum.

**Training and Sampling Hyperparameters.** Given in Table 4

|          | Property                 | Value                   |
|----------|--------------------------|-------------------------|
| **Model** | Model type              | MLP                     |
|          | Number of layers         | 3                       |
|          | Hidden dimension         | 512                     |
|          | Activation               | ReLU                    |
| **Training** | Training iterations   | 16,000 epochs           |
|          | Batch size               | 1024                    |
|          | Optimizer                | AdamW                   |
|          | Learning rate            | $1 \times 10^{-4}$      |
|          | Noise scale $\sigma$     | 4.0                     |
|          | Diffusion coefficient    | $g(t) = \sigma^t$       |
|          | SDE type                 | Variance-Exploding (VE) |
| **Sampling** | SPT chains            | 30                      |
|          | SPT integration steps    | 50                      |
|          | SPT batch size           | 1024                    |
|          | Resampling interval (FK) | 10 steps                |
|          | FK time steps            | 50                      |
|          | FK Steering particles    | 1024                    |
|          | Guidance strength $\beta$ | 20                     |
|          | Spike widths $\omega^2$  | .01 (sharp), .1 (wide)  |

*Table 4.* Training, model, and sampling configuration for the 1D Gaussian mixture toy example. Both SPT and FK Steering use the same trained model.

### B.1.2. ABLATION STUDY: GUIDANCE STRENGTH

**Experimental Settings.** We use the primary 1D toy setting (Appendix B.1.1) and vary the guidance strength $\beta$ to study the trade-off between exploration and exploitation.

**Guidance Regimes.**

- **Exploratory:** $\beta = 5$

- **Exploitative:** $\beta = 100$

**Sampling Configuration.** All other hyperparameters remain identical to Appendix B.1.1. The global minimum sharpness is fixed at $\omega^2 = 0.01$.

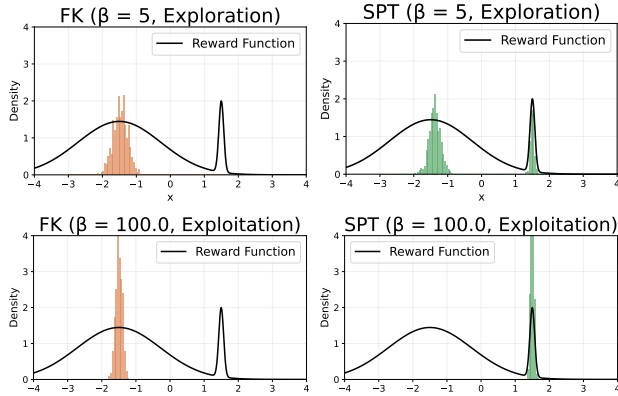

*Figure 5.* Comparison of sampling strategies on the mixture-of-Gaussians toy problem. Each panel shows 1024 samples generated by: (a) FK-Steering with low guidance strength (exploratory), (b) high guidance strength (exploitative), (c) SPT with low guidance strength (exploratory), and (d) SPT with high guidance strength (exploitative). FK Steering is dominated by the high-mass local minimum and assigns minimal probability to the global optimum, whereas our method can either explore multiple minima or concentrate on the global optimum, depending on the guidance strength. Note here we plot against the reward function for clarity.

**Reward-Adjusted Distribution.**

$$p_\beta(x) \propto p(x) \exp(\beta R(x)), \tag{49}$$

with $p(x)$ and $R(x)$ defined in Appendix B.1.1.

**Purpose.**    Varying $\beta$ emphasizes reward over likelihood, controlling the transition from broad multimodal exploration to concentrated sampling at the global reward optimum. Results are shown in Figure 5.

### B.1.3. EXTENDED TOY SETTING: TWO-DIMENSIONAL MULTIMODAL DISTRIBUTION

**Experimental Settings.**    We extend the analysis to a 2D four-component Gaussian mixture to assess the robustness of our methods under increased dimensionality and more complex multimodal structure. The hyperparameters below govern model training and sampling.

**Data Distribution.**    We extend the 1D mixture to a 2D four-component Gaussian mixture:

$$x \sim 0.45\,\mathcal{N}((-1.5, 0), 0.2^2 I) + 0.05\,\mathcal{N}((1.5, 0), 0.2^2 I) \tag{50}$$
$$+ 0.25\,\mathcal{N}((0, -1.5), 0.2^2 I) + 0.25\,\mathcal{N}((0, 1.5), 0.2^2 I) \tag{51}$$

**Model, training, and sampling configuration.**    Given in Table 5

**Reward Function.**

$$R(x) = R_{\text{global}}(x) + R_{\text{local}}(x), \tag{52}$$

$$R_{\text{global}}(x) = -\text{ReLU}\left(-\exp\left(-\frac{(x_1 - 1.5)^2 + x_2^2}{0.1}\right) \cdot 100 + 1.9\right), \tag{53}$$

$$R_{\text{local}}(x) = -\text{ReLU}\left(-\exp\left(-\frac{(x_1 + 1.5)^2 + x_2^2}{1.0}\right) \cdot 2 + 1.8\right) \tag{54}$$

The global optimum is sharp and high-reward but lies on a low-probability mode, while the local optimum is broader and aligns with the dominant data mode. The ReLU is used to *flatten the tops of the peaks*, preventing the reward from being dominated by a single point with extremely high strength. This is done to effectively sample from a single Gaussian for visualization purposes.

| | Property | Value |
|---|---|---|
| **Model** | Model type | MLP |
| | Input/output dimension | 2 |
| | Number of layers | 3 |
| | Hidden dimension | 512 |
| | Activation | ReLU |
| **Training** | Training iterations | 16,000 epochs |
| | Batch size | 1024 |
| | Optimizer | AdamW |
| | Learning rate | $1 \times 10^{-4}$ |
| | Noise scale $\sigma$ | 4.0 |
| | Diffusion coefficient | $g(t) = \sigma^t$ |
| | SDE type | Variance-Exploding (VE) |
| **Sampling** | SPT chains | 30 |
| | SPT integration steps | 50 |
| | SPT batch size | 4096 |
| | Resampling interval (FK) | 10 steps |
| | FK time steps | 50 |
| | FK Steering particles | 4096 |
| | Guidance strengths $\beta$ | 10 (exploration), 100 (exploitation) |
| | Spike width $\omega^2$ | .01 |

*Table 5.* Training, model, and sampling configuration for the 2D Gaussian mixture toy example. Both SPT and FK Steering use the same trained model.

**Purpose.**   The 2D toy experiment evaluates the robustness of our flow-based sampling methods under increased dimensionality and modal complexity. Results are shown in Figure 6.

### B.2. Dynamical System Trajectory Sampling

We follow the experimental details from [14; 23]. Notably, FK Steering requires a nondeterministic sampler, and SPT requires a deterministic sampler. Given the equivalence between diffusion and flow matching models, we train one flow matching model [29] with the variance-exploding probability path [43] and then use an Euler-Maruyama sampler for FK Steering and a Heun sampler for SPT. To reduce computational cost, we discretize the sampling interval [0, 1] into 100 points, rather than 1,000 as done by Finzi et al. [14]. Figure 7 shows that the flow matching model still satisfactorily samples $p(\boldsymbol{x})$.

Across all experiments, we fix $\beta = 8.29$ for Lorenz '63, and $\beta = 10.27$ for FitzHugh-Nagumo. The other parameters of FK Steering and SPT are tuned for each dynamical system using grid search to maximize the success rate, the fraction of 32,000 sampled trajectories that are in the event $E$. The parameters of FK Steering we consider are the ensemble size (number of particles used) in the sequential Monte-Carlo process, and the potential (difference, max, or sum); the intermediate reward function is always evaluated at $\mathbb{E}[\boldsymbol{x}_1|\boldsymbol{x}_t]$ where $\boldsymbol{x}_1$ is a denoised trajectory. Figure 8 plots a heatmap of the grid search results.

**Lorenz '63.**   The chaotic dynamical system introduced in [31] is defined as

$$\dot{\boldsymbol{x}} = \begin{bmatrix} \dot{x}_1 \\ \dot{x}_2 \\ \dot{x}_3 \end{bmatrix} = F(\boldsymbol{x}) = \begin{bmatrix} \sigma(x_2 - x_1) \\ x_1(\rho - x_3) - x_2 \\ x_1 x_2 - \beta x_3 \end{bmatrix}$$

for $\sigma, \rho, \beta \in \mathbb{R}$. Following Finzi et al. [14], we choose $\sigma = 10$, $\rho = 28$, and $\beta = 8/3$, and use a scaled system $\hat{F}(\boldsymbol{x}) = F(20\boldsymbol{x})/20$ such that $x_i(\tau) \in [-3, 3]$ for all dynamics time $\tau$ and $i \in \{1, 2, 3\}$. The event of when the trajectory stays on one arm of the chaotic attractor is characterized by $C(\boldsymbol{x}) = 0.6 - \|\mathcal{F}[\boldsymbol{x} - \bar{\boldsymbol{x}}]\|_1 > 0$ where $\mathcal{F}$ is the Fourier transform over $\tau$, $\|\cdot\|_1$ is the 1-norm over frequency magnitudes and system dimension, and $\bar{\boldsymbol{x}}$ is the mean of $\boldsymbol{x}(\tau)$ over $\tau$.

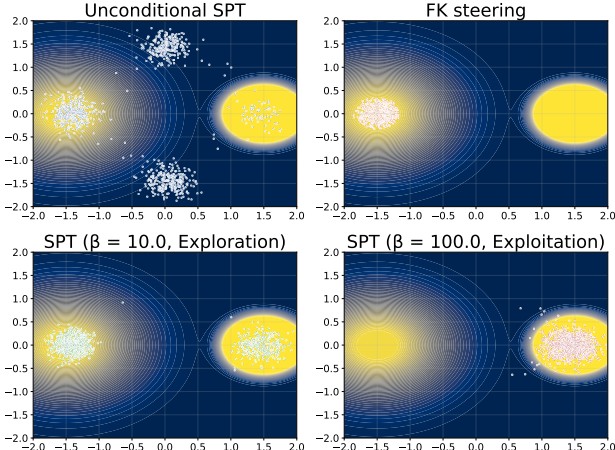

*Figure 6.* Comparison of sampling strategies on the mixture-of-Gaussians toy problem. Each panel shows 1024 samples generated by: (a) unconditional source-guided diffusion, (b) FK Steering with high guidance strength (exploitative), (c) SPT with low guidance strength (exploratory), and (d) SPT with high guidance strength (exploitative). The reward function is plotted as a contour. FK Steering is dominated by the high-mass local minimum and assigns minimal probability to the global optimum, whereas our method can either explore multiple minima or concentrate on the global optimum, depending on the guidance strength. Note for FK Steering only $\beta = 100.0$ is shown for clarity, as both values produced qualitatively similar results.

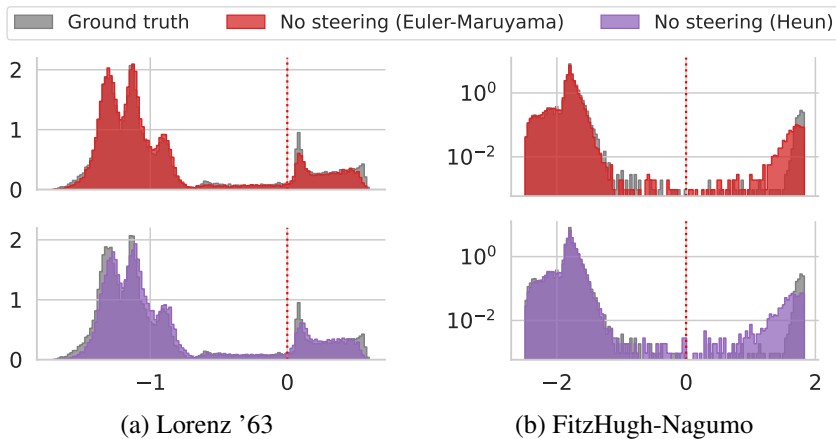

*Figure 7.* Distribution of $C(\boldsymbol{x})$ where $\boldsymbol{x}$ is sampled using Dormand-Prince and a flow matching model with 100 sampling time steps without steering.

**FitzHugh-Nagumo.** The dynamical system introduced in [15; 33] that models an excitable neuron is defined as

$$\dot{x}_i = x_i(a_i - x_i)(x_i - 1) - y_i + k \sum_{j=1}^{d} A_{ij}(x_j - x_i)$$

$$\dot{y}_i = b_i x_i - c_i y_i$$

for $i \in \{1, 2\}$. We use the parameters used in [13; 14; 23]: $a_1 = a_2 = -0.025794$, $b_1 = 0.0065$, $b_2 = 0.0135$, $c_1 = c_2 = 0.2$, $k = 0.128$, and $A_{ij} = 1 - \delta_{ij}$ where $\delta_{ij}$ is the Kronecker delta. The event of neuron spiking is characterized by $C(\boldsymbol{x}) = \max_\tau [x_1(\tau) + x_2(\tau)]/2 - 2.5 > 0$.

**Training Dataset.** We use the Dormand-Prince ODE solver [11] to generate trajectories of each dynamical system for training with $1.4 \times 10^{-8}$ and $1 \times 10^{-6}$ for the absolute and relative tolerances, respectively. The initial conditions of the trajectories are sampled from Gaussian distributions and then are evolved by the ODE solver for some number of "burn-in" time steps. For Lorenz '63, we sample $\mathcal{N}(\boldsymbol{0}, \boldsymbol{I})$ and use 30 "burn-in" time steps, and for FitzHugh-Nagumo, we sample $\mathcal{N}(\boldsymbol{0}, (0.2)^2\boldsymbol{I})$ with 250 time steps for "burn-in." The trajectories saved to the training dataset begin from the burned-in

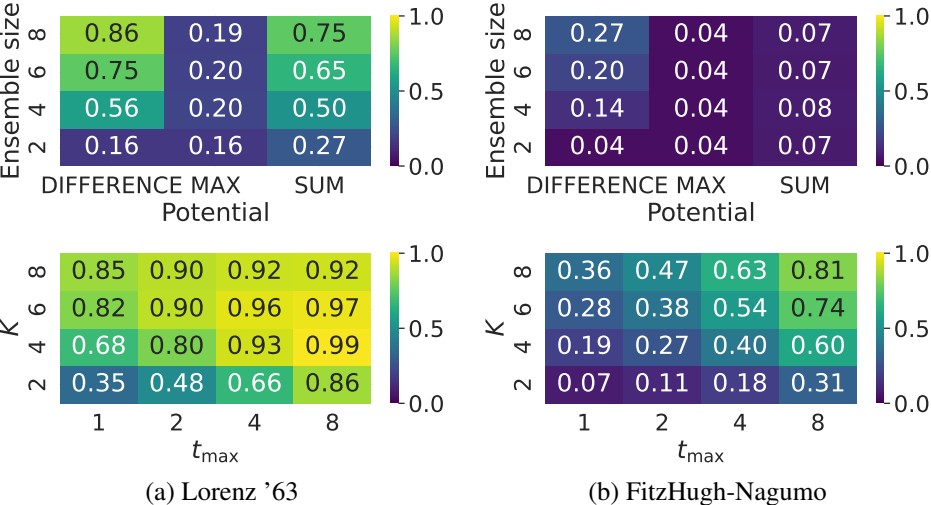

(a) Lorenz '63         (b) FitzHugh-Nagumo

*Figure 8.* Success rates from grid searches over the parameters of FK Steering (top row) and SPT (bottom row) for each dataset.

initial condition and are 60 time steps long, with a time step size of 0.1 for Lorenz '63 and 6.0 for FitzHugh-Nagumo.

**Model hyperparameters and training.** The generative model uses the UNet architecture used by [14], and is trained with the flow matching loss [29] with the variance-exploding probability path [43]. The flow matching loss is weighted by $1/(\sigma'_{1-t})^2$ where $\sigma_t$ is the variance-exploding noise standard deviation such that $\sigma_0 \approx 0$. We train it on a dataset of 4,000 trajectories with a batch size of 500 for 2,000 epochs using the Adam optimizer, an initial learning rate of $1 \times 10^{-4}$, and an exponential decay learning rate scheduler with a rate of 0.995. We evaluate the model using an exponential moving average of its parameters during training with a smoothing factor of $1/400$.

### B.3. Text-to-Image Generation

**Experimental Setting.** We largely follow the experimental protocol of FK Steering. Steering experiments use pretrained Stable Diffusion v1.4, v1.5, and XL models. Stable Diffusion v2.1 is not included because it was unavailable through HuggingFace at the time of experimentation. Results for FK Steering, base Stable Diffusion, DPO, and DDPO baselines are taken directly from the FK Steering paper.

**Reward Model.** As in FK Steering, we use the ImageReward-v1.0 human preference model as the reward function for all text-to-image steering experiments.

**Sampler Modification for SPT.** To enable Source Parallel Tempering (SPT), which requires a deterministic transport map, we modify the DDIM sampler by setting the noise parameter to $\eta = 0$ with 30 SPT iterations.

**Shared Inference Hyperparameters.** See Table 6 for details.

**Diversity Metric.** To quantify sample diversity, we use CLIP embedding diversity. Let $f_\theta$ denote the CLIP image encoder and $\{\mathbf{x}_0^i\}_{i=1}^k$ the $k$ generated samples for a prompt. We define

$$\text{CLIP-DIV}(\{\boldsymbol{x}_0^i\}_{i=1}^k) = \sum_{i=1}^k \sum_{j=i}^k \frac{2}{k(k-1)} \|f_\theta(\boldsymbol{x}_0^i) - f_\theta(\boldsymbol{x}_0^j)\|_2^2 \qquad (55)$$

Table 2 displays the average CLIP diversity across all generated samples for a given prompt.

**SPT Step Ablation** We conduct an ablation study to evaluate how the number of SPT tempering steps affects both aesthetic quality and sample diversity. Inference-time steering is performed using the ImageReward HPS model on a subset

| Category | Property | Value |
|---|---|---|
| **Base model** | Model Type | Stable Diffusion v1.4, v1.5, SDXL |
| **Sampling (Shared)** | Classifier-free guidance scale | 7.5 |
| | Batch size | 4 |
| | Maximum temperature | 10 |
| **Sampling SPT** | Type | DDIM, $\eta = 0$ (deterministic) |
| | Diffusion steps | 50 |
| | Chains | 3 |
| | SPT Iterations | 30 |
| **Sampling FK Steering** | Type | DDIM, $\eta = 1$ (stochastic) |
| | Diffusion steps | 100 |

*Table 6.* Shared inference and steering configuration for text-to-image experiments. All methods use identical diffusion and guidance settings, with SPT differing by a deterministic sampler and SPT iterations.

of 30 randomly selected prompts from the ImageReward benchmark. For each prompt, we record (i) the ImageReward score of the highest-reward sample and (ii) the CLIP embedding diversity across generated samples. All other hyperparameters are held fixed to isolate the effect of the number of SPT steps.

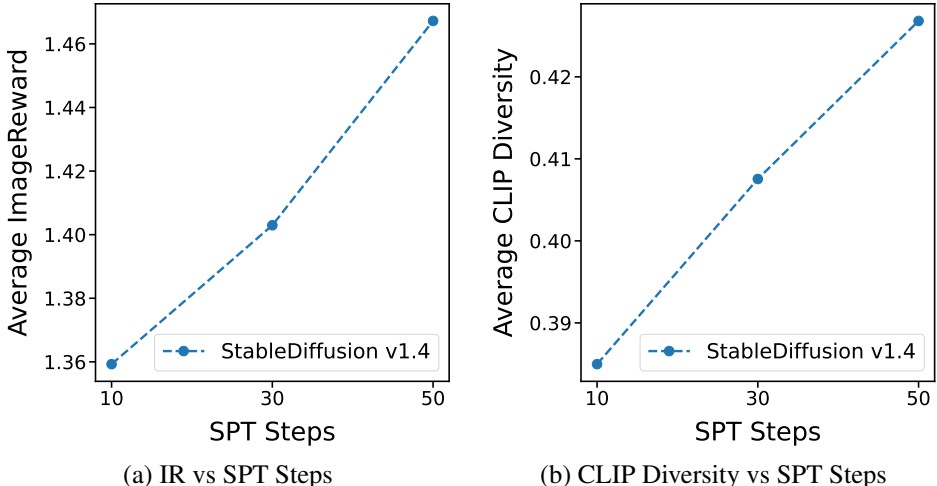

(a) IR vs SPT Steps        (b) CLIP Diversity vs SPT Steps

*Figure 9.* Average ImageReward and CLIP Diversity of generated samples when steering Stable Diffusion v1.4.

Figure 9 shows that increasing the number of SPT steps leads to consistent improvements in both reward and diversity. This suggests that additional tempering steps allow SPT to better explore the reward-tilted distribution while maintaining multiple diverse trajectories.

**Parallel Tempering Abalation**   We conduct an ablation study to evaluate the impact of the parallel tempering mechanism on generated sample quality. Inference-time steering is performed using the ImageReward HPS model on a subset of 10 randomly selected prompts from the ImageReward benchmark. To isolate the contribution of parallel tempering, we compare the standard SPT algorithm with $K = 3$ chains against a variant with a single chain ($K = 1$), which effectively removes the parallel tempering mechanism. For each prompt, we record the ImageReward score of the highest-reward sample and report the average score across all prompts. All other hyperparameters are held fixed across experiments.

Table 7 displays the results of the parallel tempering experiment. Across all tested models, incorporating parallel tempering consistently improves reward alignment, yielding higher average ImageReward scores compared to the single-chain ($K = 1$) variant.

| Model | Without Parallel Tempering | With Parallel Tempering |
|---|---|---|
| Stable Diffusion v1.4 | 0.546406 | **1.39** |
| Stable Diffusion v1.5 | 0.84849 | **1.33** |
| Stable Diffusion XL | 0.682567 | **1.27** |

*Table 7.* Comparison of ImageReward scores for Stable Diffusion v1.4, v1.5, and XL with and without parallel tempering.

### B.3.1. EFFICIENCY COMPARISON WITH FK STEERING AND SPT

**Experimental Setup.** We compare the generation quality of FK Steering and Source Parallel Tempering (SPT) for steering Stable Diffusion v1.4 using the ImageReward [49] reward function using a fixed number of function evaluations (NFE) (Fig. 3). For each prompt in the ImageReward benchmark, we generate four images under a fixed model evaluation budget of $5,250$ function evaluations. Matching the NFE ensures a fair comparison of the methods' ability to optimize for reward under equal computational cost.

**FK Steering Configuration.** FK Steering uses an ensemble size of $5$, the maximum potential function, and $1,000$ sampling time steps. It also recomputes the ensemble's potentials every 20 steps and resamples if the effective sample size (ESS) falls below $2.5$. The total NFE is therefore $5 \cdot (1,000 + 1,000/20) = 5,250$. Spending the NFE on more sampling steps provides FK Steering with more opportunities to update the ensemble based on the reward signal.

**SPT Configuration.** For SPT, we use $K = 3$ trajectories, $t_{\max} = 34$ inner optimization steps, and 50 sampling steps, giving $3 \cdot (34 + 1) \cdot 50 = 5,250$ NFE. Following [39], the tilt strength is set to $\lambda = 10 = \beta$. In contrast to FK Steering, SPT allocates its NFE for optimizing the trajectory along the deterministic transport map rather than frequent resampling.

**Fixed Budget Abalation** We also conduct an ablation study to evaluate the impact of the NFE budget on generated sample quality using Stable Diffusion v1.4. Inference-time steering is performed using the ImageReward HPS model on 10 randomly selected prompts from the ImageReward benchmark under varying NFE budgets.

To match compute budgets across methods, we adjust the ensemble size for FK Steering while keeping the inference steps and resampling schedule fixed. Specifically, for the $4,200$ NFE setting, we use $K = 4$, corresponding to $4 \cdot (1,000 + 1,000/20) = 4200$, while for the $6,300$ NFE setting we use $K = 6$, corresponding to $6 \cdot (1,000 + 1,000/20) = 6,300$.

For SPT, we instead vary the number of SPT steps while keeping the number of chains fixed at $K = 3$. For the $4,200$ NFE setting, we use $t_{\max} = 27$, yielding $3 \cdot (27 + 1) \cdot 50 = 4,200$, while for the $6,300$ NFE setting we use $t_{\max} = 41$, yielding $3 \cdot (41 + 1) \cdot 50 = 6,300$.

For each method, we compute the average reward of generated samples for each prompt and then report the average across all prompts.

| NFE Budget | 4,200 | 5,250 | 6,300 |
|---|---|---|---|
| FK Steering | 0.157 | 0.535 | 0.449 |
| SPT (Ours) | **1.161** | **1.152** | **1.192** |

*Table 8.* Comparison of ImageReward scores for FK Steering and SPT across various compute budgets.

Table 8 shows that SPT consistently achieves higher reward alignment than FK Steering across all compute budgets. Furthermore, increasing the NFE budget generally improves SPT performance, suggesting that additional computational budget enables more effective exploration of high-reward regions in the reward-tilted target distribution.

