# OpenReview forum: "Test-Time Guidance for Flow-Based Generative Models via Parallel Tempering on Source Distributions"
_ICML.cc/2026/Conference — ICML 2026 regular_

### Official Review · Reviewer_yL6z · 2026-02-19

**Soundness:** 4
**Presentation:** 4
**Significance:** 4
**Originality:** 3
**Overall Recommendation:** 5
**Confidence:** 5

**Summary:**

The paper is positioned in the context of guidance for generative models, namely the modification of the generative pipeline in order to favor specific subsets of the data distribution (e.g., particular types of images or prompt-conditioned outputs). The central idea is to define rewards in the source (latent) space rather than directly in data space, and then steer source samples so that their pushforward through the generative map yields data points with high reward.
In the theory section, the authors build upon existing results on source-guided flow matching. They extend Theorem 4.2 by replacing a strong assumption based on an $L_\infty$ bound with a weaker condition involving bounded squared loss. Under this relaxed assumption, they prove that in the limit of zero approximation error of the generative map, the sampled distribution coincides with the desired tilted distribution.
In the algorithmic section, the authors introduce Parallel Tempering in source space to efficiently generate samples associated with high reward after pushforward. The method is first tested on a simple one-dimensional Gaussian mixture example, chosen to clearly highlight the differences with a state-of-the-art gradient-free baseline (FK steering).
Finally, the method is benchmarked against both FK steering and DDPO (a reinforcement-learning-based approach) on real-world tasks, including image generation with Stable Diffusion and protein generation. Across several evaluation metrics, the proposed approach outperforms the competing methods.

**Compliance With Llm Reviewing Policy:**

Affirmed.

**Final Justification:**

My concerns have been adequately addressed, hence I keep my score as Accept.

**Key Questions For Authors:**

These are relatively minor points that I would encourage the authors to address in the final version.
1. Related recent work

After a brief search, I found two very recent works that appear potentially relevant:
  - Carter, A., Choi, S., Tamogashev, K., Elvira, V., & Malkin, N. (2026). Discrete diffusion samplers and bridges: Off-policy algorithms and applications in latent spaces. arXiv:2602.05961.
  - Kalaivanan, A., Zhao, Z., Sjölund, J., & Lindsten, F. (2025). ESS-Flow: Training-free guidance of flow-based models as inference in source space. arXiv:2510.05849.

It would be beneficial for the authors to clarify to what extent these works are related or conceptually connected to the present contribution. Even if the overlap is limited, a brief discussion situating the present method relative to these approaches would strengthen the positioning of the paper and appropriately acknowledge adjacent developments, without diminishing the originality of the current work.

2. Performance degradation with respect to \omega in FK steering

Regarding the comparison with FK steering (Figure 1), is there a theoretical reason to expect performance degradation as a function of \omega? If so, this point could be emphasized more clearly in the discussion. Providing either an intuitive or formal explanation would help the reader interpret the empirical trends more deeply, rather than viewing them as purely experimental observations.

3. Interpretation of protein generation metrics

In the protein experiments (Table 3), the proposed method appears to position itself between FK steering and ADP-3D in terms of ELBO and RMSD. However, from a biological perspective, it would be helpful to better contextualize these metrics. For readers who are not domain experts, ELBO and RMSD in isolation can be difficult to interpret.
I would suggest adding a short explanation of what constitutes a “good” ELBO or RMSD in this setting, and possibly including a qualitative or visual example illustrating structural differences. Otherwise, one might simply conclude that the proposed method performs “in between” competitors, without understanding whether this corresponds to a meaningful biological trade-off or improvement.

4. Limitations

The limitations of the work are briefly mentioned at the end, under future perspectives. I would encourage the authors to elaborate slightly more on this aspect. A clearer discussion of potential weaknesses, computational bottlenecks, scalability issues, or assumptions (e.g., dependence on the quality of the generative map approximation) would further strengthen the paper and provide a balanced view of the contribution.

**Limitations:**

As for the comment above, I suggest to the author to put in the final version a more detailed list of limitations of the present work.

**Strengths And Weaknesses:**

Regarding the four axes of evaluation — soundness, presentation, significance, and originality — I have very few criticisms to raise.

Overall, the paper is, in my opinion, very sound. The core idea is clearly articulated from the outset and remains coherent throughout the manuscript. The theoretical section is concise yet sufficiently detailed, and it is transparent how the authors build upon prior results. This clarity makes the logical progression of the work easy to follow and enhances the reader’s understanding of the development of the main contributions.

The presentation is also very well structured. Apart from minor comments (listed below), the manuscript is clear and well organized. I found the paper particularly easy to read, which is not always the case in interdisciplinary areas such as generative modeling and advanced sampling methods. The experimental section is also well designed, progressing naturally from simple illustrative examples to more realistic image-generation benchmarks, in line with current state-of-the-art practices in generative modeling.

The work is, in my view, significant. It explores novel directions in the emerging field of source-guided generative modeling, both at the theoretical and algorithmic levels. In particular, the introduction of Parallel Tempering — a well-established state-of-the-art technique in MCMC sampling — into the source-guidance framework is conceptually elegant and practically meaningful. This combination provides a principled way to efficiently generate high-reward source samples prior to pushforward.

---

> ### Author Rebuttal · Authors · 2026-03-30
>
> We thank the reviewer for the thoughtful and constructive feedback. We are particularly grateful for your very positive assessment of our work and for endorsing our submission. In what follows, we provide detailed point-by-point responses to the comments and questions raised.
>
> ---
>
> **Q1. Clarify how Carter, A. et al, 2026 and Kalaivanan, A. et al., 2025 are related to the present contribution.**
>
> **Response:** Thank you for bringing up these two relevant works. We have expanded our related work section in the revision to include the following discussion:
>
> ESS‑Flow (Kalaivanan et al., 2025): While this work shares our perspective on source-space inference, it focuses on a single-temperature, importance-sampling-style correction. Consequently, it does not address multimodality or barrier crossing. In contrast, SPT explicitly targets these challenging regimes using parallel tempering, which is crucial for navigating sharply peaked reward landscapes.
>
> Carter et al. (2026) study discrete diffusion and off‑policy bridge methods in latent spaces, focusing on correctness in discrete settings. Their discrete setting fundamentally differs from our focus on reward-tilted test-time guidance for pretrained flows on continuous space.
>
> ---
>
> **Q2. Regarding the comparison with FK steering (Fig 1), is there a theoretical reason to expect performance degradation as a function of $\omega$?**
>
> **Response:**  As $\omega$ decreases, the reward becomes increasingly sharp and concentrated in regions of vanishing probability within the base distribution, which fundamentally conflicts with the mechanics of FK steering. FK steering relies on sequential importance weighting and resampling along the generation trajectory; when the reward peak is narrow, almost all particles receive negligible weights until extremely late in the process, causing resampling to collapse onto base-distribution modes rather than crossing low-density barriers. Thus, the observed degradation with smaller $\omega$ reflects a known failure mode of FK–type particle systems under low-overlap target distributions with respect to the base distribution. In contrast, SPT avoids this collapse by performing MCMC directly in source space and using parallel tempering to maintain overlap across temperatures, which preserves mixing even as $\omega \to 0$. We will emphasize this explanation more clearly in the revision.
>
> ---
>
> **Q3. In the protein experiments (Table 3), SPT appears to position itself between FK steering and ADP-3D in terms of ELBO and RMSD. However, from a biological perspective, it would be helpful to better contextualize these metrics. ....**
>
> **Response:** We appreciate these suggestions.
>
> RMSD measures structural similarity between backbone conformations in Ångströms (Å), where lower values indicate closer agreement with a reference structure. ELBO captures how plausible a generated protein is under the Chroma generative prior, where higher values reflect greater structural realism. For context, results for a similar in-silico experiment (ESS Flow) with Chroma span roughly 11.4–17.0 Å (RMSD) and -8.1–8.9 (ELBO).
>
> Together, these metrics capture two complementary aspects of generation quality: how structurally consistent the output is with known conformations, and how plausible it is as a protein in its own right. A method that scores well on both is preferable, but when the two are in tension, we argue that structural plausibility is the more consequential criterion for downstream utility. For scientific applications, a generated protein that is physically unrealizable offers limited value regardless of its RMSD.
>
> Under this framing, SPT's performance is stronger than the "in between" characterization suggests. Its RMSD is comparable to competing methods, indicating consistent structural agreement, while its ELBO of 10.01–10.25 sits entirely outside the range of all competing methods (-8.1–8.9)—a qualitatively distinct gain in structural plausibility. The two metrics together therefore tell a coherent story: SPT matches competitors on structural consistency while meaningfully exceeding them on realism, which we would argue constitutes a genuine biological improvement rather than a simple trade-off.
>
> ---
>
> **Q4. I would encourage the authors to elaborate slightly more on the limitation of the work.**
>
> **Response:** Thank you for this suggestion—we agree that a more explicit limitation discussion would improve the balance and clarity of the paper. We will discuss the computational cost, and emphasize SPT’s dependence on the quality of the learned generative map: while our theory shows robustness under standard $L^2$ training error, large approximation errors can still degrade guidance performance. Additionally, SPT introduces algorithmic hyperparameters (e.g., temperature ladders, proposal scales, swap frequencies) whose tuning is problem-dependent.
>
> ---
>
> We sincerely appreciate your efforts in reviewing our manuscript and for considering our rebuttal.

---

> > ### Author Rebuttal · Reviewer_yL6z · 2026-03-31
> >
> > Thanks for the response. I keep my score. I just suggest that a part in the appendix, maybe also visual, is added regarding my point of Q3. For instance to show how a good ELBO and terrible RMSD looks like and viceversa compared to some ground true. I find very important, for large venues, that a visual help for general readers is added if possible.

---

> > > ### Author Response · Authors · 2026-03-31
> > >
> > > We are grateful for your prompt response and continued support of our work. As you suggested, we will incorporate this visual into the appendix to illustrate the contrast between a well‑calibrated ELBO with poor RMSD and the converse, anchored by ground‑truth examples. We agree that this addition will make the contribution more accessible to a broader audience at large venues.

---

### Official Review · Reviewer_hwyz · 2026-03-04

**Soundness:** 3
**Presentation:** 3
**Significance:** 3
**Originality:** 3
**Overall Recommendation:** 4
**Confidence:** 2

**Summary:**

Generative model have proven successful in generating data samples from a simple, known source distribution. However, it is still very common to use guided generation processes to ensure that specific constrains are met. Existing steering methods fail to get out of local minima, need gradients and additionally miss theoretical justification. The authors therefore propose Source Parallel Tempering (SPT) a novel gradient-free guidance framework, that operates entirely in the source space. By combining parallel tempering and pCN Operator, for steering the source distribution toward a high-reward region. The paper additionally underpins their approach theoretically. SPT shows superior performance on 3 different task, image synthesis, protein structure and dynamical systems.

**Compliance With Llm Reviewing Policy:**

Affirmed.

**Final Justification:**

The paper represents a novel and well-motivated idea, with a theoretical grounding and demonstrates applicability in a wide range of tasks: image synthesis, protein structure and dynamical systems. The authors address my concerns regarding RMSD/ELBO trade-off and hyperparameter sensitivity. My main concern regarding the structure of the first chapters, was promised to be addressed in the revised version. While the paper needs some minor polishing, the experimental validation and the soundness of the contributions have encouraged me to raise my score.

**Key Questions For Authors:**

1. Can you comment on the computational overhead of SPT and how the computation budget is measured? How well does the method scale ?
2. All image generation experiments use Stable Diffusion variants with the ImageReward human preference model as the reward. How does SPT perform when the reward is adversarial or poorly calibrated i.e., when maximizing R(x) leads to reward hacking artifacts? FK Steering's resampling collapse might actually be protective in such cases.
3.  The authors mention extending the work to Large Language Models (LLMs) or discrete diffusion as a future path. What improvements do you expect from the method?

**Limitations:**

yes

**Strengths And Weaknesses:**

## **Strengths**
  * The paper introduces a novel error bound linking $L_2$ training error directly to test-time guidance quality ($W_2$ distance), addressing gaps in prior $L_{\infty}$ based theory.
  * Through SPT, guidance can be applied in gradient free regimes and it can be used for high dimensional problem settings, making it applicable to a wide range of problem settings.
  * SPT show stable results, achieving up to $50\%$ improvement over baselines.
  * SPT is tested on a variety of problem settings and demonstrates superior performance in all of these

## Weaknesses
* For the protein generation results, SPT achieves excellent ELBO scores, its RMSD is substantially worse than ADP-3D. The paper frames it as if SPT balances these metrics but the gap is very large, and one may question whether SPT actually solves the protein conditioning task or simply avoids reward overfitting while failing to match the target geometry.
* As mentioned in Section 5.1. (Local Exploration) the effectiveness of the tempering ladder depends on the choice of the number of chains and temperature increments, which can be intensive to tune.
* Methodological presentation of the paper would need  improvement. No clear structure with the numberings. Additional related work section does not contribute to increase clarity and would better fit in the appendix or should be included in the introduction.
* Figure 1 (c) is missing a description

---

> ### Author Rebuttal · Authors · 2026-03-30
>
> We thank the reviewer for the thoughtful and constructive feedback. We are particularly grateful for your recognition of the novel error bound, the broad applicability, and the superior performance of our proposed approach. In what follows, we provide detailed point-by-point responses to the comments and questions raised.
>
> ---
>
> **W1. SPT achieves strong ELBO scores but substantially worse RMSD than ADP-3D. Does SPT actually solve the protein conditioning task, or does it merely avoid reward overfitting while failing to match target geometry?**
>
>
> **Response:** We thank you for raising this point. We agree this gap deserves clearer interpretation and will revise the manuscript accordingly. While ADP‑3D outperforms SPT on geometric matching, SPT targets a different failure mode in protein conditioning. ADP-3D achieves lower RMSD by aggressively optimizing geometry at the cost of substantial ELBO degradation—indicative of leaving the learned structural prior and producing physically implausible conformations. SPT instead constrains guidance to remain close to the pretrained distribution, limiting RMSD reduction but preventing pathological structures. From a biological perspective, this reflects a meaningful tradeoff: SPT prioritizes generating plausible proteins that move toward the target geometry, rather than perfectly matching the target at the cost of realism.
>
> ---
>
> **W2. As mentioned in Section 5.1, the effectiveness of the tempering ladder depends on the choice of the number of chains and temperature increments, which can be intensive to tune.**
>
>
> **Response:** We thank the reviewer for raising this concern. In our experiments, uniform temperature increments were sufficient in all tested settings, showing that tuning the number of chains and SPT iterations is sufficient to improve performance. We refer the reviewer to Figure 9, Appendix B.2 for an ablation study over these parameters.
>
> ---
>
> **W3. Methodological presentation of the paper would need improvement. No clear structure with the numberings. Additional related work section does not contribute to increase clarity and would better fit in the appendix or should be included in the introduction.**
>
> **Response:** We appreciate your suggestions. We have added a subsection to the introduction outlining the paper's organization to help readers navigate the paper. We have also moved the additional related work section to the introduction.
>
> ---
>
> **W4. Figure 1 (c) is missing a description.**
>
> **Response:** Thank you for pointing this out. We have added the description $W_2$ distance vs. $\omega^2$ in the revised paper.
>
> ---
>
> **Q1. Can you comment on the computational overhead of SPT and how the computation budget is measured? How well does the method scale?**
>
> **Response:** The computational budget is measured in NFE given by $t_{\max} \times K \times$  inference_steps, scaling linearly with the number of SPT steps, chains, and steps per transport evaluation. Regarding computational overhead, with the same NFE budget (i.e., essentially the same computational time), SPT attains substantially higher ImageReward than FK Steering on the text-to-image task (see table in our response to Q1, Q4, & W3 of Reviewer BSWB).
>
> ---
>
> **Q2. All image generation experiments use Stable Diffusion variants with the ImageReward human preference model as the reward. How does SPT perform when the reward is adversarial or poorly calibrated i.e., when maximizing R(x) leads to reward hacking artifacts? FK steering's resampling collapse might actually be protective in such cases.**
>
> **Response:** Thank you for this insightful comment. SPT is explicitly designed to faithfully sample from the reward‑tilted distribution. Handling adversarial or poorly calibrated reward functions is an important and interesting problem, however, it is beyond the scope of our current work.
>
> ---
>
> **Q3. The authors mention extending the work to Large Language Models (LLMs) or discrete diffusion as a future path. What improvements do you expect from the method?**
>
> **Response:** We have not dove into LLM or discrete diffusion yet. Below, we provide some of our initial thoughts. We expect the idea of SPT can improve LLMs and discrete diffusion models when the reward function is non-differentiable or exhibits regions of low-probability but high-reward. For instance, for discrete diffusion models, operating in a learned source/latent token space would allow test‑time guidance to explore alternative generations without committing early to irreversible decoding decisions, while parallel tempering offers a principled mechanism for escaping local optima induced by greedy or single‑temperature reward shaping. Compared to existing approaches such as best‑of‑K sampling or rejec­tion‑based reranking, we expect improved coverage of valid high‑reward modes and better robustness to multimodal rewards.
>
> ---
>
> We sincerely appreciate your efforts in reviewing our manuscript and for considering our rebuttal.

---

> > ### Author Rebuttal · Reviewer_hwyz · 2026-04-02
> >
> > The questions have been addressed and I will raise my score.

---

> > > ### Author Response · Authors · 2026-04-02
> > >
> > > Thank you again for considering our rebuttal. We are glad that we have addressed all of your concerns regarding our submission, and we appreciate your willingness to increase the score of our paper.
> > >
> > > We noticed that the updated score has not yet appeared in the system, and we would be very grateful if you could update it at your convenience. Thank you very much for your time and support.

---

### Official Review · Reviewer_BSWB · 2026-03-08

**Soundness:** 3
**Presentation:** 2
**Significance:** 3
**Originality:** 3
**Overall Recommendation:** 5
**Confidence:** 3

**Summary:**

The authors propose Source Parallel Tempering (SPT), a novel steering technique for test-time guidance of flow-based models. They argue that, unlike previous test-time steering techniques, their approach avoids complex data-space exploration and does not require tractable gradients. SPT uses Markov chain Monte Carlo in the source (initial noise) space of flow-based models to sample noise vectors that are biased toward higher-reward outputs. The authors run experiments on text-to-image models, protein generation models, and dynamical system trajectory sampling. They report significant improvements on selected benchmarks across all three settings.

**Compliance With Llm Reviewing Policy:**

Affirmed.

**Final Justification:**

My concerns were fully addressed. I raise my score to an "Accept".

**Key Questions For Authors:**

1. Please address the points in "Strength and Weaknesses"
2. Could you show that the strong results come from parallel tempering itself, rather than simply from spending more inference-time compute on search?
3. In Figure 4, you report ImageReward scores for SPT and FK-Steering under a fixed budget of 5250 evaluations. Why was that number chosen, and how do the results change under different fixed compute budgets?
4. In the text-to-image experiments, how exactly is the “highest quality sample” selected? Do the conclusions still hold when reporting average quality or quality under a fixed compute budget?

**Limitations:**

yes

**Strengths And Weaknesses:**

**Strengths
1. The paper's targeted problem is relevant: test-time steering of pretrained generative models, especially in settings where the reward is not differentiable or hard to optimize.
2. The core idea is clear and well motivated: steer the well-behaved initial noise space instead of the more complex intermediate spaces or the data space.
3. The approach is gradient-free, which broadens its applicability beyond settings where reward gradients are available.
4. The authors run experiments on several, diverse domains, including text-to-image generation, protein generation, and dynamical system trajectory sampling.

**Weaknesses:
1. The comparison to baselines is limited. The authors mainly compare against DPO and FK Steering. DPO is a training-time alignment method, so the main relevant comparison is FK Steering. The paper lacks comparisons to simpler inference-time baselines, such as a best-of-K approach (e.g. [1]).
2. The main theoretical claims are presented informally in the paper body, which weakens the presentation.
3. Reporting the “highest quality sample generated per prompt” favors search-heavy methods, and it is unclear how exactly the best sample is selected.

[1] Karthik, Shyamgopal, et al. "If at first you don't succeed, try, try again: Faithful diffusion-based text-to-image generation by selection." arXiv preprint arXiv:2305.13308 (2023).

---

> ### Author Rebuttal · Authors · 2026-03-30
>
> We thank the reviewer for the thoughtful review and valuable feedback, as well as for commending the novelty of our methods and the significance of our experimental results. In what follows, we provide point-by-point responses to the comments and questions raised.
>
> ---
>
> **Q1 & W1. The comparison to baselines is limited, e.g., lacking comparison over best-of-K.**
>
> **Response:** We appreciate your suggestion. Following FK steering, we have included the best-of-K baseline for the text-to-image experiment. Best-of-K is a meaningful baseline here because the text prompt provides conditional guidance.
>
> |Model|1.4|1.5|XL
> |-|-|-|-|
> |Best-of-K|0.800|0.737|1.236|
> |SPT (ours)|1.409|1.411|1.479|
>
> For our other tasks, however, best-of-K is uninformative. Because those tasks lack conditional guidance, best-of-K merely evaluates the prior distribution's natural overlap with the reward function, rather than measuring the effectiveness of guided generation.
>
> ---
>
> **Q1 & W2. The main theoretical claims are presented informally in the paper body.**
>
> **Response:** We appreciate the reviewer's comment. Our goal in the main text was to prioritize readability and highlight our core insights without compromising their implications. Therefore, we chose to present simplified theoretical statements in the main body. The fully rigorous statements, which explicitly define all constants using standard theoretical constants (e.g., the Lipschitz constants of the reward function and transport map), are deferred to the appendix. This structure is designed to ensure readers can easily grasp the main takeaways without being distracted by dense technical details.
>
> ---
>
> **Q1, Q4, & W3. Reporting the “highest quality sample generated per prompt” favors search-heavy methods. In text-to-image, how exactly is the “highest quality sample” selected? Do the conclusions still hold when reporting average quality or quality under a fixed compute budget?**
>
> **Response:** For each prompt, we define the highest-quality sample as the one in the coldest chain with the highest reward. Following FK steering, Table 1 reports the average of these highest-reward samples over all ImageReward prompts.
>
> To compare average sample quality under fixed compute budgets, we designed a new experiment on a subset of ImageReward prompts. For each method, we computed the per-prompt average reward and then averaged over prompts. We report results, in the table below, across three compute budgets and find that SPT's average sample quality consistently exceeds that of FK steering. We have included this table in the revision.
>
> |NFE budget|4200|5250|6300|
> |-|-|-|-|
> |FK-steering|0.157|0.535|0.449|
> |SPT|1.161|1.152|1.192|
>
> ---
>
> **Q2. Could you show that the strong results come from parallel tempering itself, rather than simply from spending more inference-time compute on search?**
>
> **Response:** Thank you for this insightful question. To isolate the effect of parallel tempering, we repeated our text-to-image experiment (Section 6.2, Table 1) on a subset of 10 prompts using only a single chain (K=1), directly removing the parallel tempering mechanism. Comparing these results to our existing (K=3) results demonstrates that parallel tempering substantially improves overall generation quality. We have included this ablation in the revision (see table below):
>
> |Model|W/O PT| W PT|
> |:-|-:|-:|
> |v1.4|0.546406|1.39|
> |v1.5|0.84849 |1.33|
> |xl|0.682567 |1.27|
>
> ---
>
> **Q3. In Figure 4, you report ImageReward scores for SPT and FK-steering under a fixed budget of 5250 evaluations. Why was that number chosen, and how do the results change under different fixed compute budgets?**
>
> **Response:** We thank the reviewer for the opportunity to clarify. To construct a comparable FK steering configuration, we have to balance the number of particles, denoising steps, and resampling opportunities. Since FK steering operates over a single set of particles, we allowed 1000 denoising steps per particle with 50 opportunities to resample, much more than SPT. With 5 particles and 50 resampling steps, this yields $5\times1000 + 5\times50 = 5250$ NFEs, a budget SPT can match cleanly with 35 steps, 3 chains, and 50 denoising steps ($35\times3\times50 = 5250$). The NFE budget of 5250, therefore, represents a practical operating point at which both methods are well within their effective range, making the comparison most informative about each method's generative quality.
>
> Additionally, we repeated the experiment on a subset of prompts at NFE budgets of 4200 and 6300, measuring average sample quality at each level. SPT showed consistent improvement over FK steering across all three budgets, suggesting the comparison is robust to the precise budget choice. Results and details on how average quality was computed can be found in our response to “Q1, Q4 & W3”.
>
> ---
>
> We sincerely appreciate your efforts in reviewing our manuscript and for considering our rebuttal.

---

> > ### Author Rebuttal · Reviewer_BSWB · 2026-04-01
> >
> > Dear Authors,
> >
> > thank you for your detailed response: my concerns have mostly been addressed. However, I remain partly unconvinced by your answer to **Q1 and W1**, specifically by your claim that Best-of-K is an uninformative baseline for unconditional models.
> > Your argue that:
> > > _"[best-of-K] merely evaluates the prior distribution's natural overlap with the reward function, rather than measuring the effectiveness of guided generation"_
> >
> > I do not find this argument convincing. The issue is not what Best-of-K evaluates or measures, but whether it is a relevant practical baseline for the setting considered in the paper. At a high level, your method takes a frozen, pretrained generative model and spends additional test-time compute to obtain samples with higher reward. Best-of-K fits exactly this same description: it also uses extra test-time compute to obtain higher-reward samples from the same fixed model. In fact, it is the most naive and direct instantiation of this idea, and for that reason it is a necessary baseline.
> >
> > Whether the pretrained model is conditional or unconditional does not change this point. In both cases, the practical question is the same: given a fixed model and a reward function, does SPT outperform simply drawing more samples and selecting the best ones?

---

> > > ### Author Response · Authors · 2026-04-02
> > >
> > > We thank the reviewer for the prompt response and continued engagement with our work. We appreciate your clarification and agree that including Best-of-K provides a valuable and practical point of comparison for test-time compute methods across other benchmarks.
> > >
> > > We have incorporated Best-of-K results for the analytic example (Section 6.1) and dynamical systems (Section 6.4) in this response, as well as in the revised manuscript. For the analytical example, we use $K=30$ (corresponding to 30 chains used in FK steering and SPT), a batch size of 1024, and 50 denoising steps. These settings are chosen to match the experimental configuration used in FK steering and SPT, ensuring a fair comparison. For dynamical systems, we use $K = 500$ and then report the success rate and total variation distance as in the main paper.
> > >
> > > **Analytic Example**
> > > | Method          	| W₂ Distance  |
> > > |-----------------------------|------------------|
> > > | FK steering     	| 2.98   	|
> > > | Best-of-K 	| 1.839  	|
> > > | SPT (Ours)      	| 0.08   	|
> > >
> > > **Dynamical Systems**
> > > |System|Method|Success Rate|Total Variation|
> > > |-|-|-|-|
> > > |Lorenz ‘63|Best-of-K|0.19|0.81|
> > > ||FK steering|0.68|0.36|
> > > ||SPT|0.99|0.14|
> > > |FitzHugh-Nagumo|Best-of-K|0.04|0.98|
> > > ||FK steering|0.04|0.97|
> > > ||SPT|0.81|0.64|
> > >
> > > In both the analytic example and dynamical systems, as well as in the previously reported image tasks, SPT consistently outperforms Best-of-K. This indicates that the observed gains from SPT go well beyond what can be achieved by simply drawing more samples from the prior.
> > >
> > > For protein design (Section 6.3), the computational scale required to match the configurations used in our existing benchmarks results in a longer runtime. We are actively working to complete these experiments for inclusion in the final version and are happy to discuss this further or provide additional details if helpful.
> > >
> > > Thank you once again for your review and for taking the time to consider our rebuttal.

---

### Official Review · Reviewer_RMKp · 2026-03-17

**Soundness:** 3
**Presentation:** 3
**Significance:** 3
**Originality:** 3
**Overall Recommendation:** 5
**Confidence:** 3

**Summary:**

This paper proposes a new method for test-time guidance in pretrained generative models. Instead of altering the learned ODE transport, it keeps that map fixed and changes the source distribution so that transporting from it yields the desired tilted target distribution. To sample from this new source distribution, the authors use parallel tempering. The paper also includes a theoretical analysis of the resulting error and a fairly broad set of experiments across different application domains.

**Compliance With Llm Reviewing Policy:**

Affirmed.

**Final Justification:**

I think the paper provides a meaningful contribution, and the authors addressed all of my questions during the rebuttal without raising any further concerns on my end. Although the core idea is not especially novel, the paper is well executed and I think it should be accepted.

**Key Questions For Authors:**

1) What exactly should the reader take away from Theorem 4.3 and Corollary 5.4 in practice?

2) Can the authors compare more directly, experimentally or at least in discussion, to other recent parallel tempering papers in ML, especially CREPE?

3) How sensitive is SPT to the temperature ladder and number of chains?

**Limitations:**

Yes

**Strengths And Weaknesses:**

Originality: This clearly builds on prior work in source-space guidance and parallel tempering, but the specific choice to keep the learned transport fixed and do reward-based tempering in source space seems novel as far as I can tell.

Significance: The paper studies an important problem, namely flexible test-time guidance for pretrained generative models, especially when the reward is not differentiable.

Presentation: I found the paper generally easy to follow and the method section was clear, though the theory took up a fair amount of space and, for me, did not add much beyond the main intuition.

Soundness: The algorithm itself makes sense to me and is the strongest part of the paper. Working in source space is a sensible design choice, the local pCN update matches the Gaussian source assumption well, and replica exchange is a natural way to move between separated high-reward regions when the reward landscape is rough or multimodal. The theory also seems directionally reasonable, and I did not notice anything obviously wrong, but I personally did not feel it gave me much beyond the basic story that if the learned transport is good and the sampler mixes well enough, then the guided samples should be good. For example, Theorem 4.3 mainly controls the error from replacing the true transport with the learned one, but it does not really speak to broader model mismatch, so I did not come away with much extra insight from it. On the empirical side, the experiments are fairly broad and generally support the main claims.

---

> ### Author Rebuttal · Authors · 2026-03-30
>
> We thank the reviewer for the thoughtful review and valuable feedback, as well as for recognizing the novelty and significance of our work. We are particularly grateful for your very positive assessment of our work and for endorsing our submission. In what follows, we provide point-by-point responses to the comments and questions raised.
>
> ---
>
> **Q1. What exactly should the reader take away from Theorem 4.3 and Corollary 5.4 in practice?**
>
> **Response:** We thank the reviewer for this question. We have revised Sections 4.2 and 5.2 to clarify the practical implications and key insights of these results.
>
> The key contribution of Theorem 4.3 is to establish a direct, training-aligned guarantee: the $W_2$ error of the generated distribution is controlled by the same $L_2$ objective used to train the transport map. While the high-level intuition (“better transport ⇒ better samples”) is natural, prior formulations (e.g., Theorem 4.2) involve mismatched error terms ($L_\infty$) that are not directly optimized in practice. Our result removes this gap, showing that standard training objectives are sufficient to control generation quality. On the other hand, Theorem 5.2 provides a $W_2$ bound on sampling error for parallel tempering, in contrast to prior work, which typically relies on total variation bounds. Both theorems hold under mild assumptions on the source distribution, while extending to more general distributions may require additional tools. Corollary 5.4 straightforwardly combines these results to provide a $W_2$ guarantee over the full pipeline. Note that $W_2$ is the standard metric for transport problems in generative models.
>
> ---
>
> **Q2. Can the authors compare more directly, experimentally or at least in discussion, to other recent parallel tempering papers in ML, especially CREPE?**
>
> **Response:** We thank the reviewer for highlighting CREPE, a highly relevant recent application of parallel tempering. While both our approach and CREPE utilize replica exchange, they differ fundamentally in their mechanics. Our method performs source steering by annealing directly from a tractable source distribution $q$ to the tilted source distribution $\tilde{q}$, which explicitly helps particles escape local modes in the target landscape. In contrast, CREPE anneals along the temporal direction (i.e., across diffusion noise levels). Furthermore, our source-steering approach is supported by rigorous theoretical bounds, whereas CREPE does not currently provide theoretical guarantees.
>
> Experimentally, a direct quantitative comparison is challenging. CREPE is not primarily focused on reward-tilted tasks. While they do include one such experiment on ImageNet, their method simultaneously adopts Classifier-Free Guidance (CFG). This makes it impossible to isolate the specific performance gains of their parallel tempering technique from the effects of CFG. Additionally, the paper only reports visual plots in this task rather than explicit numerical rewards, precluding a direct numerical baseline comparison.
>
> We have included a discussion of these methodological differences in the revised manuscript.
>
> ---
>
> **Q3. How sensitive is SPT to the temperature ladder and number of chains?**
>
> **Response:** Thank you for this insightful question. We refer the reviewer to the ablation study in Figure 9 (Appendix B.2), which varies the number of SPT steps and the number of chains, and reports the corresponding success rates. We also include ablations for FK steering with different ensemble sizes and potential functions for comparison.
> Overall, the results show that **SPT is robust across a wide range of hyperparameter choices and consistently outperforms FK steering**:  For Lorenz ’63, when $K \geq 4$, for FitzHugh–Nagumo, across all cases.
>
> ---
>
> We sincerely appreciate your efforts in reviewing our manuscript and for considering our rebuttal.

---

> > ### Author Rebuttal · Reviewer_RMKp · 2026-04-02
> >
> > Thank you for your rebuttal. I maintain my positive score.

---

> > > ### Author Response · Authors · 2026-04-02
> > >
> > > We are grateful for your prompt response and continued support of our work.

---

### Decision · Program_Chairs · 2026-04-30

**Decision:**

Accept (regular)

**Comment:**

This paper introduces source parallel tempering for reward alignment of diffusion/flow-based generative models. The idea is interesting, as performing parallel tempering over the Gaussian source is quite an appealing idea, as it allows you to pass high-energy barriers. Overall, the motivation of the paper is clear, and the image experiments show consistent gains over FK-steering, which is useful. I am, however, torn on this paper as the protein experiments with Chroma as the base model are abysmally bad for RMSD. In fact, I would recommend removing this experiment, because an RMSD >10 is a useless structure---to the point where the method is not working at all. Please note that ELBO is also not a useful metric. There are plenty of appropriate design metrics, such as designability (which currently fails), TM-score-based diversity/novelty. These are now quite standard and significantly more appropriate here. The authors should be much more forthright with respect to the limitations of these experiments if they decide to keep this in the paper---or even remove this wholesale. Concretely, an expanded discussion is warranted, with some qualitative analysis of the generated structures.

Despite this, the freshness of the method and strong image experiments is enough to warrant acceptance.